# MUTATION2TEXT: A UNIFIED PROTEIN AND TEXT LANGUAGE MODEL FOR EXPLAINING MUTATION EFFECTS

## ABSTRACT

Understanding the functional consequences of protein mutations is crucial for diagnosing and preventing diseases like cancer. However, existing protein language models (PLMs) are limited by their black-box nature, inability to process full-length proteins without truncation, and primary focus on single-nucleotide variants. We introduce **Mutation2Text**, a multimodal generative PLM designed to generate human-understandable, rationale-based explanations for diverse mutations, including substitutions, insertions, deletions, and frameshifts. Its architecture uses a gated cross-attention mechanism to explicitly contrast wild-type and mutated sequences and a Perceiver Resampler for length-invariant encoding. We constructed Mutation2TextQA, the largest mutation interpretation dataset to date, comprising millions of question-answer pairs with substantial lexical and semantic diversity, mined from published literature, facilitating robust generalization across mutation contexts. Mutation2Text sets a new state-of-the-art across benchmarks for mutation explanation, pathogenicity prediction, and disease association. Critically, our analysis uncovers a significant discrepancy between the model's highly accurate internal representations (0.96 AUC on pathogenicity) and its final text output (0.85 AUC), highlighting a key "lost-in-translation" challenge for generative AI in science. We release all code, data, and models to facilitate future research at `https://anonymous.4open.science/r/mutation2text-94A6/`.

## 1 INTRODUCTION

Understanding how genetic mutations alter protein function is a critical challenge in medicine (Cooper & Shendure, 2011; Yang et al., 2013). Mutations in genes can disrupt signaling pathways, molecular interactions, or structural stability, leading to diseases ranging from rare disorders to common cancers. For the majority of observed mutations (also called variants), their clinical impact remains uncertain (Richards et al., 2015). The prevalence of these variants of uncertain significance (VUS) is staggering. ClinVar (the primary variant repository) reports that 47% of its 5.4 million entries are VUS (Fowler & Rehm, 2024); most tumor genomic sequencing reports include at least one somatic VUS (Mellgard et al., 2024); and of the $\sim 4$ million individuals sequenced globally each year, 1–2 million receive results containing germline VUS (Chen et al., 2023), leaving them without actionable guidance on surveillance or treatment.

While protein language models (PLMs) are powerful for learning sequence representations (Rives et al., 2021; Lin et al., 2022b; Nijkamp et al., 2023), they are not inherently mutation-aware. Trained on wild-type sequences from diverse organisms, they typically assess a variant's impact using zero-shot log-likelihoods—a proxy for how evolutionarily unlikely the mutation is (Cheng et al., 2023). However, many pathogenic variants, particularly those causing late-onset human diseases that don't affect reproduction, are not constrained by evolutionary pressure and thus appear permissible to a PLM (Hou et al., 2025; Meier et al., 2021) does not reproduction. As a result, standard PLMs can struggle on this critical class of disease-causing variants, particularly for late-onset phenotypes where evolutionary constraint is weak, motivating architectures that reason directly over mutation context and clinical evidence.

Current LLMs that map full-length protein sequences to text (Appendix A.1) either lack the ability to reason on mutated sequences, or lack the generality to operate directly on full-length protein sequences, compare residue-level changes, and produce human-interpretable explanations across diverse variant types. To overcome these limitations, we introduce **Mutation2Text**, a mutation-aware generative architecture that jointly encodes and *contrasts* wild-type and mutated protein sequences—without truncation– to generate residue-level rationales across diverse variant types. To support this, we construct **Mutation2TextQA**, a large-scale (to our knowledge, largest) mutation-focused dataset mined from PubMed, and leverage functional, interaction, motif, and post-translational context of the wild-type protein to infer mutational impact.

**Our work makes the following contributions.** (i) A mutation-aware generative model that operates directly on protein sequences and explicitly contrasts wild-type vs. mutant residues. (ii) Broad support for complex variant classes (SNVs, indels, multi-site mutations, frameshifts) without truncation artifacts. (iii) A gated cross-attention + Perceiver design that scales to long proteins and enables rich contextual reasoning. (iv) We perform extensive experiments showing that Mutation2Text advances the state of the art on mutation explanation, disease prediction, and text-based pathogenicity prediction, and we provide a detailed analysis of a "lost-in-translation" gap between highly predictive latent embeddings and less precise natural-language explanations.

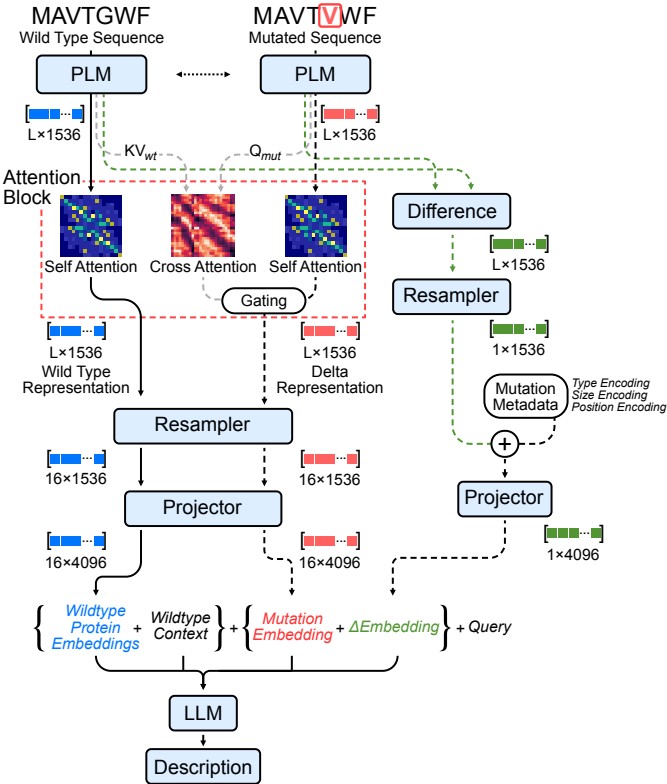

Figure 1: Overview of the Mutation2Text architecture. The model encodes both wild-type and mutated protein sequences using a protein language model, with an Attention block that learns using both embeddings. A Perceiver resampler and Projection layer reduce number of embedding tokens and project from protein space to text space respectively. In addition, a mutation feature extraction module computes vector capturing mutation type, size, and position, which is projected into the same space as other embeddings to form the *delta embedding*. The user query and wildtype context (function, motifs, interactions, PTMs) are tokenized and passed as input to the large language model together with wild-type embeddings, mutated embeddings, and delta embedding enabling the LLM to generate natural language explanations of the mutation.

## 2 METHODS

The **Mutation2Text** architecture, associated multi-modal input preparation and Dataset COllection and Processing for training, evaluation, and inference are as follows.

### 2.1 MODEL ARCHITECTURE

The mutation-aware generative framework, **Mutation2Text** (Figure 1), comprises six principal components: Protein Language Model (**PLM**), **Attention Block**, **Perceiver Resampler**, **Projector**, **Delta-Mutation Features Path**, and **LLM**. The **PLM** is employed to extract representative features from protein sequences, while the **Attention Block** is specifically designed to capture mutation-aware interactions between wild-type and mutant protein features. The **Perceiver Resampler** transforms variable-length protein representations into fixed-length embeddings. The **Delta-Mutation Features Path**,highlighted with green data flow, encodes information derived from sequence differences, mutation type, size, and positional metadata to provide delta embeddings. The **Projector** subsequently aligns both the fixed-length protein embeddings and the mutation-difference embeddings with the natural language embeddings corresponding to the human query. Finally, the instruction-tuned **LLM** component (LLaMA-3.1-8B) integrates the wild-type context, query, wild-type embeddings, mutant embeddings, and delta embeddings to generate natural language explanations of the mutation.

The data flow and other implementation details of the model's functional components are outlined as follows:

**Protein Language Model (PLM) and The Attention Block (Figure 1)**: A single instance of ESM-3 (`esm3_sm_open_v1`) (Hayes et al., 2025) is fine-tuned to encode wild-type and mutated protein sequences. Final hidden-layer embeddings serve as keys, queries, and values for the **Attention Block**, which enables reasoning between wild-type and mutant features. It operates under three modes depending on whether the input contains only the wild-type, only the mutant, or both sequences (Appendix A.3). Components include: (i) *Shared Self-Attention Layer*, where a *single multi-head self-attention* processes both representations (Appendix A.3.1); (ii) *Cross-Attention Layer*, where mutant embeddings attend to wild-type context (Appendix A.3.2); and (iii) *Gating Mechanism*, which adaptively fuses outputs from self- and cross-attention (Appendix A.3.3).

**Perceiver Resampler (Figure 1)**: Since **Attention Block** embeddings vary with sequence length, the **Perceiver Resampler** employs an algorithm inspired by Flamingo (Alayrac et al., 2022) to produce fixed-length embeddings. Given $L$ sequence embeddings, it outputs a 16-length representation (Appendix B).

**MLP Projector (Figure 1)**: Fixed-length embeddings from the wild-type and mutant resamplers—shown in blue and red, respectively—are mapped to the LLM input space via a two-layer **MLP**, which projects each of the 16 latent vectors from 1536 to 4096 dimensions (Appendix C).

**Delta-Mutation Features Path(Figure 1)**  : Depicted in green, this pathway runs parallel to the protein feature stream and captures mutation-specific information beyond sequence differences. It incorporates delta embeddings together with structured metadata such as mutation type (substitution, insertion, deletion, frameshift), size (change in length), and normalized position along the sequence, producing a single mutation feature vector that is passed to the LLM along with the other embeddings (Appendix D).

The Attention Block and Delta-Mutation Features Path play complementary roles. The gated cross-attention module provides a comparison between wild-type and mutant features, allowing the mutant representation to attend to relevant residues in the wild-type sequence. In parallel, the delta path encodes "hard" priors about the mutation, its type, size, position, and local ESM-based differences, into a single token that serves as a structural anchor. Table 5 show that removing the delta token degrades performance, supporting this design choice.

### 2.2 MULTIMODAL INPUT PREPARATION FOR LLM

Protein representations are combined with textual input by tokenizing the *wild-type context*—which describes the protein's function, structural motifs, molecular interactions, and PTMs—alongside the

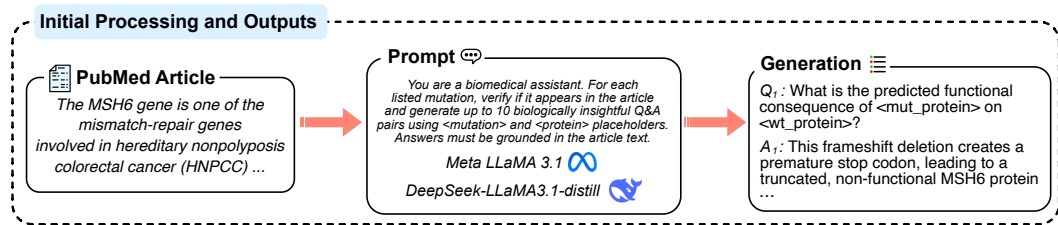

Figure 2: Pipeline for Mutation2TextQA Generation.

user query using the LLaMA tokenizer. Each token $x_i$ is mapped to an embedding via the standard embedding layer, producing $\texttt{TokenEmbed}(x_i)$ for every token. These token embeddings constitute the primary input stream, into which additional embeddings are inserted at specified marker positions.

The three types of embeddings inserted via special tokens in the input are defined as follows:

- $\texttt{<wt\_protein>}$ and $\texttt{</wt\_protein>}$: The wild-type projected embedding $\mathbf{E}_{\text{protein\_wt}} \in \mathbb{R}^{16 \times 4096}$ is inserted between these markers.
- $\texttt{<mut\_protein>}$ and $\texttt{</mut\_protein>}$: The mutant projected embedding $\mathbf{E}_{\text{protein\_mut}} \in \mathbb{R}^{16 \times 4096}$ is inserted between these markers.
- $\texttt{<mut\_features>}$: a single mutation feature token (delta embedding) $\mathbf{E}_{\text{mut\_feat}} \in \mathbb{R}^{1 \times 4096}$ derived from mutation metadata (type, size, position) and sequence differences.

Specifically, (i) the projected wild-type protein embedding $\mathbf{E}_{\text{protein\_wt}} \in \mathbb{R}^{16 \times 4096}$, (ii) the projected mutant embedding $\mathbf{E}_{\text{protein\_mut}} \in \mathbb{R}^{16 \times 4096}$, and (iii) a mutation feature token $\mathbf{E}_{\Delta\text{mut}} \in \mathbb{R}^{1 \times 4096}$, are added with wild-type textual context and query tokens. The final multi-modal input sequence can be summarized as:

$$\mathbf{E}_{\text{input}} = \mathbf{E}_{\text{protein\_wt}} + \mathbf{E}_{\text{wt\_context}} + \mathbf{E}_{\text{protein\_mut}} + \mathbf{E}_{\Delta\text{mut}} + \mathbf{E}_{\text{query}}.$$

This unified sequence $\mathbf{E}_{\text{input}} \in \mathbb{R}^{T' \times 4096}$ is passed to the LLM for generation (Algorithm 2).

## 2.3 DATASET COLLECTION AND PROCESSING

### 2.3.1 MUTATION2TEXTQA

The **Mutation2TextQA** dataset curation was done with a pipeline consisting of four stages: (1) mutation–article pairing via ClinVar (Landrum et al., 2016), (2) QA generation using a large language model (LLM), (3) protein sequence resolution, and (4) homology-aware train–test splitting.

**Mutation–Article Pairing via ClinVar** :ClinVar's $\texttt{var\_citation}$ table (Landrum et al., 2016) provides mappings between variant identifiers and supporting PubMed article IDs. For each mutation entry (identified by $\texttt{VariationID}$, $\texttt{rs}$, or $\texttt{AlleleID}$), the associated PubMed citations are extracted.

**QA Generation via DeepSeek**: For each PubMed article ID and its mutation mentions, DeepSeek-Chat (DeepSeek-AI et al., 2025) was used to generate 10 QA pairs per mutation. As shown in Table 8, the prompt instructs the model to (i) verify whether the mutation is discussed in the article; (ii) generate biologically meaningful questions using $\texttt{<mutation>}$ and $\texttt{<protein>}$ placeholders (questions only); and (iii) provide article-grounded, answerable explanations without hallucinations or speculation.

**Protein Sequence Resolution with test- train split** :Wild-type and mutant amino acid sequences are obtained via the Mutalyzer API (Lefter et al., 2021). For each transcript-level mutation (HGVS coding notation), the wild-type reference sequence ($\texttt{RefSeq}$) is retrieved and modified to produce the mutant sequence ($\texttt{MutSeq}$). Entries lacking complete sequence pairs, missing annotations, or

unresolvable mutations are excluded. The remaining data are split into train/test sets as detailed in Appendix E.

### 2.3.2 CLINVARQA

**ClinVarQA**,constructed for this work, is a large-scale dataset of templated question–answer pairs derived from the ClinVar (Landrum et al., 2016) variant summary table, filtered for pathogenic or benign coding variants. Each entry generates 6–10 QA pairs using handcrafted templates (Table 15), with answers populated directly from metadata fields such as `phenotype`, `type`, and `pathogenicity`.

While Mutation2TextQA provides free-form, literature-derived question–answer pairs about functional effects, ClinVarQA focuses on templated questions derived from structured ClinVar fields (e.g., clinical significance, disease labels).

### 2.3.3 GLOBAL HOMOLOGY-AWARE SPLIT ACROSS DATASETS

To reduce train–test leakage via closely related proteins, we use a single global homology-aware split across ClinVarQA and Mutation2TextQA, rather than splitting each dataset independently. We first pool all unique wild-type protein sequences across the two sources and compute pairwise sequence identities using global alignment from BioPython library. We then build a similarity graph where nodes are sequences and edges connect pairs whose identity exceeds a fixed threshold. Connected components of this graph define sequence clusters. Finally, we assign clusters, not individual examples, to train (80%) or test (20%) such that all entries associated with a given wild-type sequence cluster (ClinVarQA pairs, and Mutation2TextQA pairs) are always placed in the same split. Test sequences are further categorized into Easy, Medium, and Hard tiers based on their maximum identity to any training sequence (e.g., $\geq 0.95$, 0.5–0.95, $< 0.5$), and all results are reported stratified by these homology split.

### 2.3.4 MUTATION2TEXTQA QUALITY CONTROL AND DEDUPLICATION

Because Mutation2TextQA is generated with the assistance of a teacher LLM, we apply explicit quality-control and deduplication steps. For each candidate QA pair and its source PubMed article, a second, independent LLM verifies three conditions: (i) the answer is directly supported by the article, (ii) the question is answerable from the article alone, and (iii) all proteins and mutations mentioned in the QA appear in the article. We retain only QA pairs for which all three checks are affirmative, discarding approximately 3.3% of generated pairs (Appendix F). Remaining errors are categorized into misinterpretation, contradiction, exaggeration, use of external knowledge, and other rare types; we summarize this hallucination taxonomy in Table 10. To avoid inflating the dataset with near-duplicate text, we represent Mutation2TextQA as a one-to-many structure analogous to image captioning benchmarks such as MS-COCO. Each entry contains a mutation, a question, and a set of distinct reference answers mined from the same article. We deduplicate exact (question, answer) strings at the mutation–article level, so that multiple legitimate rationales for the same mutation are preserved as multiple references at evaluation time, rather than as separate training examples.

## 3 EXPERIMENTS AND RESULTS FOR MUTATION INTERPRETATION

In this section, we evaluate Mutation2Text across a diverse set of variant interpretation tasks to assess its generalization capabilities. A key aspect of our experimental design is that we train a single, unified model and evaluate it across all tasks without any task-specific fine-tuning. This approach is chosen not only for computational efficiency but also to rigorously test the model's foundational reasoning abilities. We begin with a description of our training strategy, followed by detailed evaluations on core tasks, starting with mutation effect explanation. Protein-only interpretation is summarized separately (Section 4).

**Training Strategy** Our training process proceeds in two distinct stages to incrementally build the model's capabilities, using the AdamW optimizer with a learning rate of $1.0 \times 10^{-5}$ with warm-up (Appendix K).

Table 1: **Performance evaluation for mutation explanation on the test sets of MutaDescribe.**
R-L: ROUGE-L. BL-2: BLEU-2.

| Model | Easy | | Medium | | Hard | | Average | |
|---|---|---|---|---|---|---|---|---|
| | R-L | BL-2 | R-L | BL-2 | R-L | BL-2 | R-L | BL-2 |
| ProLLaMA (Lv et al., 2024) | 1.02 | 0.64 | 1.00 | 0.91 | 1.03 | 0.70 | 1.02 | 0.74 |
| Mol-Instructions (Fang et al., 2024) | 5.10 | 0.65 | 5.19 | 0.65 | 5.56 | 0.90 | 5.27 | 0.73 |
| Galactica-6.7B (Taylor et al., 2022) | 6.53 | 3.52 | 7.64 | 3.58 | 7.33 | 2.88 | 7.13 | 3.33 |
| GPT-4-0613 (1-shot) (OpenAI et al., 2023) | 8.04 | 2.93 | 9.96 | 3.42 | 9.62 | 2.69 | 9.14 | 3.00 |
| GPT-4-0613 (5-shot) (OpenAI et al., 2023) | 10.46 | 2.51 | 10.31 | 2.81 | 10.79 | 1.88 | 10.52 | 2.40 |
| GPT-4-0613 (5-shot, kNN) (OpenAI et al., 2023) | 11.63 | 9.63 | 12.98 | 10.88 | 12.46 | 8.63 | 12.31 | 9.69 |
| GPT-4 + ESM-2 (Lin et al., 2022a) | 11.69 | 11.09 | 13.02 | 11.50 | 12.77 | 8.48 | 12.45 | 10.37 |
| GPT-4 + OntoProtein (Zhang et al., 2022) | 11.84 | 10.93 | 12.69 | 11.22 | 12.81 | 8.17 | 12.42 | 10.13 |
| AugmentedESM (Hsu et al., 2022) | 11.60 | 8.33 | 11.40 | 7.46 | 10.73 | 6.95 | 11.26 | 7.62 |
| Fine-tuned ESM-2 (Lin et al., 2022a) | 20.49 | 9.37 | 11.87 | 5.95 | 11.34 | 3.32 | 14.88 | 6.36 |
| MutaPLM (Luo et al., 2024) | **25.80** | **18.77** | 21.07 | 12.59 | 16.51 | 8.69 | 21.34 | 13.61 |
| **Mutation2Text (Ours)** | 24.30 | 17.92 | **23.13** | **13.51** | **20.28** | **9.96** | **22.57** | **13.79** |

Table 2: Comparison of model performance on Disease Prediction and Pathogenicity Prediction
tasks. Best results per metric/difficulty are in bold.

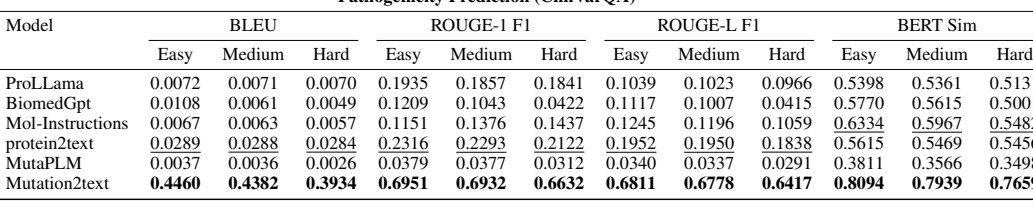

| **Disease Prediction (ClinVarQA)** | | | | | | | | | | | | |
|---|---|---|---|---|---|---|---|---|---|---|---|---|
| Model | BLEU | | | ROUGE-1 F1 | | | ROUGE-L F1 | | | BERT Sim | | |
| | Easy | Medium | Hard | Easy | Medium | Hard | Easy | Medium | Hard | Easy | Medium | Hard |
| ProLLama | 0.0068 | 0.0067 | 0.0060 | 0.1264 | 0.1032 | 0.1225 | 0.1045 | 0.0883 | 0.1007 | 0.4373 | 0.3733 | 0.4080 |
| BiomedGpt | 0.0075 | 0.0074 | 0.0070 | 0.1114 | 0.0689 | 0.0606 | 0.0922 | 0.0655 | 0.0559 | 0.4647 | 0.4517 | 0.4169 |
| Mol-Instructions | 0.0186 | 0.0172 | 0.0129 | 0.1320 | 0.1227 | 0.1063 | 0.1048 | 0.0995 | 0.0881 | 0.6165 | 0.5768 | 0.5863 |
| protein2text | 0.0157 | 0.0139 | 0.0135 | 0.1664 | 0.1563 | 0.1594 | 0.1298 | 0.1210 | 0.1183 | 0.6163 | **0.6179** | **0.5964** |
| MutaPLM | 0.0069 | 0.0057 | 0.0052 | 0.0753 | 0.0742 | 0.0669 | 0.0647 | 0.0650 | 0.0581 | 0.4890 | 0.4698 | 0.4411 |
| Mutation2text | **0.1900** | **0.1832** | **0.1146** | **0.3514** | **0.3329** | **0.1616** | **0.3244** | **0.3056** | **0.2514** | **0.6433** | 0.6171 | **0.6213** |
| **Pathogenicity Prediction (ClinVarQA)** | | | | | | | | | | | | |
| Model | BLEU | | | ROUGE-1 F1 | | | ROUGE-L F1 | | | BERT Sim | | |
| | Easy | Medium | Hard | Easy | Medium | Hard | Easy | Medium | Hard | Easy | Medium | Hard |
| ProLLama | 0.0072 | 0.0071 | 0.0070 | 0.1935 | 0.1857 | 0.1841 | 0.1039 | 0.1023 | 0.0966 | 0.5398 | 0.5361 | 0.5131 |
| BiomedGpt | 0.0108 | 0.0061 | 0.0049 | 0.1209 | 0.1043 | 0.0422 | 0.1117 | 0.1007 | 0.0415 | 0.5770 | 0.5615 | 0.5001 |
| Mol-Instructions | 0.0067 | 0.0063 | 0.0057 | 0.1151 | 0.1376 | 0.1437 | 0.1245 | 0.1196 | 0.1059 | 0.6334 | 0.5967 | 0.5482 |
| protein2text | 0.0289 | 0.0288 | 0.0284 | 0.2316 | 0.2293 | 0.2122 | 0.1952 | 0.1950 | 0.1838 | 0.5615 | 0.5469 | 0.5456 |
| MutaPLM | 0.0037 | 0.0036 | 0.0026 | 0.0379 | 0.0377 | 0.0312 | 0.0340 | 0.0337 | 0.0291 | 0.3811 | 0.3566 | 0.3498 |
| Mutation2text | **0.4460** | **0.4382** | **0.3934** | **0.6951** | **0.6932** | **0.6632** | **0.6811** | **0.6778** | **0.6417** | **0.8094** | **0.7939** | **0.7659** |

**Stage 1: Foundational protein-function grounding** : To instill a foundational knowledge of protein function, we first train only the adapter modules (Attention Block, Perceiver Resampler, and MLP Projector) while keeping the ESM-3 and LLaMA-3.1 backbones frozen. This stage uses a dataset of approximately 390,000 protein sequences and their corresponding functional summaries from UniProt. This modality-bridging step is trained for one epoch with a global batch size of 120. **Stage 2: Mutation-specific reasoning** : we fine-tune the ESM-3 encoder and the LLaMA-3.1 backbone using Low-Rank Adaptation (LoRA) alongside the adapters. Training utilizes a composite dataset of three mutation-focused sources: our ClinVar-QA and Mutation2TextQA datasets, and the MutaDescribe dataset. This full-model training proceeds for 17680 steps. The entire two-stage training process takes approximately 7 days on 8 NVIDIA Blackwell B200 GPUs. Additional details and hyperparameter settings are presented in Appendix K.

**Baselines** For a comprehensive comparison, we evaluate against two main categories of models (details in Appendix H). The primary baseline is MutaPLM , a specifically trained for mutation interpretation using MutaDescribe . The second category includes a broad range of general and protein-focused LLMs evaluated under zero-shot or few-shot prompting conditions, including text-based models like GPT-4, and specialized models ESM-2, OntoProtein, Galactica-6.7B, and Mol-Instructions.

Table 3: Overall results comparing models on Protein2text-QA benchmark. Best results per column are shown in **bold**.

| Model | BLEU-2 | BLEU-4 | ROUGE-1 | ROUGE-2 | ROUGE-L | METEOR | BERTScore | BioMedBERT |
|---|---|---|---|---|---|---|---|---|
| BioMedGPT* | 0.075 | 0.0347 | 0.159 | 0.0536 | 0.1429 | 0.139 | 0.750 | **0.905** |
| Mol-Instructions* | 0.067 | 0.038 | 0.193 | 0.0953 | 0.172 | 0.282 | 0.744 | 0.880 |
| ProtT3* | $7 \times 10^{-6}$ | $9 \times 10^{-7}$ | 0.062 | 0.001 | 0.061 | 0.017 | 0.769 | 0.843 |
| Protein2Text | 0.043 | 0.0248 | 0.265 | **0.148** | **0.239** | **0.326** | **0.815** | 0.897 |
| Mutation2Text | **0.0959** | **0.0467** | **0.2737** | 0.1023 | 0.2244 | 0.1950 | 0.6139 | 0.4684 |

## 3.1 TASK 1: MUTATION EFFECT EXPLANATION

Our first evaluation task assesses the model's ability to generate human-readable explanations of a mutation's impact on protein function, benchmarked on the MutaDescribe dataset —a benchmark for generating textual explanations of known variants.

**Findings.** On MutaDescribe, Mutation2Text delivers the strongest performance on the more challenging splits. As shown in Table 1, the baseline MutaPLM is competitive on the easy split (ROUGE-L 25.80), but Mutation2Text outperform it on medium (23.13 vs. 21.07) and hard (20.28 vs. 16.51). Mutation-agnostic and general-purpose models perform markedly worse. MutaPLM was trained solely on MutaDescribe for 200k steps; its sharp drop from easy to medium/hard is consistent with overfitting or memorization to the training distribution. By contrast, Mutation2Text, trained on larger and more diverse corpora with multi-task objectives, benefits from additional supervision that acts as a data-dependent regularizer, supplying an inductive bias across tasks and datasets and thereby reducing overfitting (Baxter, 2000).

**Beyond lexical overlap.** Because mutation explanations are long and multi-fact, surface metrics such as BLEU, ROUGE, and BERTScore are imperfect proxies for biomedical correctness. We therefore complement these metrics with a tripartite factuality protocol. First, a blinded PhD-level domain expert scores model outputs on the MutaDescribe test set as Correct, Partially Correct, or Incorrect based solely on biological factuality. Mutation2Text increases the proportion of correct or partially correct explanations from 1% for MutaPLM to 38% (10% correct, 28% partial). Second, we use two independent LLM judges to assign 0–5 factuality scores, and observe strong agreement between the judges and significant correlation with human labels (Appendix G ). Mutation2Text again outperforms MutaPLM (2.58 vs. 1.8 average LLM score). Third, we report CIDEr, which emphasizes rare, informative phrases: Mutation2Text achieves 0.499 vs. 0.013 for MutaPLM. In the remainder of the paper we treat BLEU/ROUGE/BERTScore as secondary lexical indicators, and rely on expert/LLM factuality scores, CIDEr, and AUC (for binary tasks) as primary measures of correctness.

## 3.2 TASK 2: PATHOGENECITY PREDICTION

We evaluate text-based pathogenicity prediction on a held-out ClinVar split, where models must generate explanations whose implied clinical significance (benign vs. pathogenic) is then mapped to binary labels. Because this is an intrinsically binary decision, we report ROC/AUC as the primary metric and treat BLEU/ROUGE only as secondary indicators. We compare Mutation2Text against specialized pathogenicity predictors (AlphaMissense [ref] and MutaPLM [ref]) as well as general LLM baselines on the same train–test split. For missense substitutions, Mutation2Text's embedding- level classifier matches or slightly improves upon AlphaMissense, and the text-based classifier is competitive. Unlike these baselines, Mutation2Text extends naturally to deletions, insertions, duplications, and frameshifts, where AlphaMissense Cheng et al. (2023) and MutaPLM Luo et al. (2024) are undefined; in these classes Mutation2Text provides both scalar scores and rationales. We further analyze pathogenicity AUC by mutation type and protein length in Section 7.

## 3.3 TASK 3: DISEASE PREDICTION

Linking a genetic variant to a specific disease is a crucial area of research. The task is ill-suited for standard classification due to the thousands of diseases and their complex interrelationships,

Table 4: **Qualitative analysis.** Comparing responses of Mutation2Text with baseline models. Correct answers are in blue and incorrect ones in red.

| Question (ID) | Ground Truth | Model Outputs |
|---|---|---|
| **Q1:** What is the inheritance pattern of `<mut_protein>`? | The mutation was identified as homozygous, with both parents being heterozygous carriers, indicating an autosomal recessive mode of inheritance. | **Mutation2Text:** The mutations follow an autosomal recessive pattern, requiring biallelic pathogenic variants for disease manifestation. 
 ProLLama: MAESFVEGDRNPPVF.... 
 BioMedGPT: May cause oculodentodigital dysplasia 
 Mol-Instructions: a single nucleotide polymorphism (SNP) in exon 2 of ARHGEF16." |

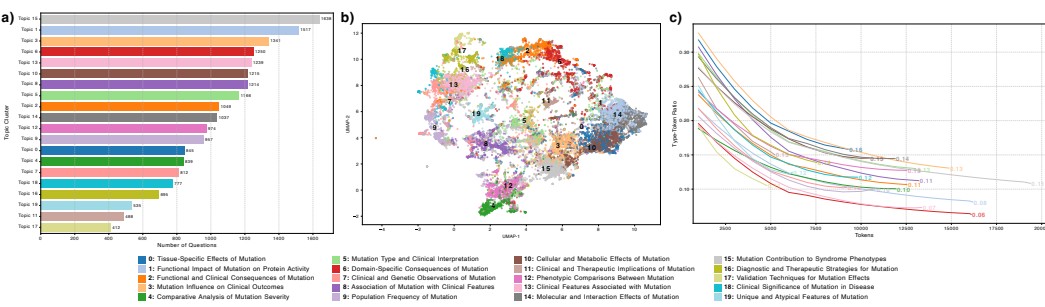

Figure 3: **Diversity Analysis of Mutation2TextQA.** a) The distribution of questions across topics from Mutation2TextQA. b) Scatter plot shows the questions in different topics. c) Lexical diversity across different topics: Type–Token Ratio vs. tokens.

making a generative approach more appropriate. We, therefore, assessed Mutation2Text's ability to generate the correct disease association from sequence information alone. Mutation2Text delivers the strongest performance, achieving the highest scores across BLEU, ROUGE, and BERT metrics (Table 2), demonstrating its capacity to map sequence changes to specific clinical outcomes.

## 3.4 ABLATION STUDY

To isolate design choices under fixed compute, we trained ablations on 10% of the data and toggled three factors: (1) contrastive inputs (Mut-only vs. WT+Mut embeddings), (2) the delta embedding (present vs. absent), and (3) sequence aggregation (Perceiver Resampler vs. mean pooling).

**Interpretation.** As show in Table 5, explicitly contrasting wild-type with mutant provides the strongest signal. The mutation-feature token adds a small gain over removing it. Using mutant context alone underperforms WT+Mut, confirming the value of retaining wild-type information. Replacing the Perceiver with mean pooling collapses performance, highlighting the importance of learned long-sequence resampling over naïve aggregation. Thus, removing any of WT-Mut contrast, mutation feature token, and Perceiver-based long-context integration degrades performance.

## 4 EVALUATION OF GENERALIZABILITY ON PROTEIN INTERPRETATION

**Experimental Setup** Although **Mutation2Text** is designed primarily for mutation reasoning, its architecture (resampler with gated cross-attention over full-length sequences) also supports protein–only function inference. To validate that Mutation2Text has learned a robust and general mapping from protein sequence to function, we evaluated it against

Table 5: ROUGE-L Scores for Ablation Study (mean ± std).

| Model Configuration | ROUGE-L |
|---|---|
| WT+Mut Embedding | $0.19 \pm 0.0071$ |
| *w/* delta embedding | $0.16 \pm 0.0058$ |
| *w/o* delta embedding | $0.15 \pm 0.0055$ |
| Mut-Only Embedding | $0.14 \pm 0.0055$ |
| *w/* Mean Pooling | $0.06 \pm 0.0036$ |

two baseline groups: (i) general-purpose LLMs (GPT-4o-mini, LLaMA-3.1-8B), and (ii) protein-specific LLMs (Mol-Instructions, BioMedGPT, Protein2Text, ProtT3) (Fang et al., 2024; Luo et al., 2023; Jararweh et al., 2025; Liu et al., 2024) . We report lexical metrics (BLEU, ROUGE, METEOR) and semantic similarity (BERTScore and BioMed-BERT similarity).Baseline descriptions and additional results appear in Appendix J.

**Results** Table 3 summarizes held-out performance. General-purpose LLMs perform poorly, consistent with the inability to interpret protein sequences. Protein-specific LLMs fare better showing semantic alignment. **Mutation2Text** achieves competitive—and often best or second-best—scores across both lexical and semantic metrics, despite protein-only interpretation not being its primary objective. This result is particularly noteworthy because Mutation2Text was not fine-tuned on the Protein2Text 14,107 dataset, making this a zero-shot evaluation of its capabilities. We also observe robustness on long proteins, aligning with the model's full-length, non-truncating design.

These results indicate that the same mechanisms that enable residue-level mutation reasoning—explicit sequence grounding and long-context encoding—also yield strong zero-shot performance for protein interpretation, which provides a solid foundation for the model's primary and more complex task of explaining mutational effect.

## 5 MUTATION2TEXTQA DATASET ANALYSIS

Mutation2TextQA contains free-form QA pairs about protein mutations extracted from the PubMed. For completeness, we breifly present here assessmen topical and semantic diversity—critical for training a model to reason across contexts. See Appendix L for details. We randomly sampled 20,000 questions, embedded them with BioMedBERT, averaged token embeddings to form question vectors, clustered with k-means (k=20), and used UMAP for visualization. For each cluster, we selected 15 nearest questions to the centroid and derived topic labels with GPT-4, followed by manual verification.

**Findings.** Figure 3a summarizes topic prevalence across clusters. The embedding space shows clear, dense clusters with limited overlap, indicating well-separated themes rather than a single dominant topic (Figure 3b). Topics span multiple biological scales—structural motifs and active-site chemistry, post-translational modification and protein–protein interaction changes, pathway rewiring, organism/tissue context, phenotype and disease associations—supporting both mechanistic and clinical reasoning. Adjacent clusters capture related subthemes (e.g., structure→function→phenotype cascades), suggesting the dataset covers transitional concepts rather than isolated silos. Figure 3c plots the dataset's type-token ratio (TTR). Lower curves in the plot indicate more repetition in the vocabulary as opposed to the top curves. Thus, Mutation2TextQA provides broad, semantically coherent coverage of mutation-centric questions with meaningful topical structure.

## 6 DISCREPANCY BETWEEN LATENT KNOWLEDGE AND TEXTUAL OUTPUT

A critical question for generative models in science is whether the final text output faithfully represents the model's internal, learned knowledge. To investigate this, we designed an experiment to separately probe the predictive power of Mutation2Text's latent embeddings versus its generated text on a 20% held-out portion of the ClinVar dataset (Section 2.3.2). First, we tested the quality of the internal representations that are fed into the LLM. We extracted these latent embeddings and used them to train a simple MLP classifier for pathogenicity prediction. Strikingly, this classifier achieved an AUC of 0.96, a performance that surpasses even highly specialized models like AlphaMissense (Figure 4). This result demonstrates that our core architecture learns a powerful and highly discriminative representation of mutation effects.

Next, we assessed whether this high-fidelity signal was successfully translated into the final output. We evaluated the generated text from Mutation2Text on the same pathogenicity task, classifying outputs based on keywords like "pathogenic" and "benign." In stark contrast to the embedding per-

Table 6: AUC by mutation type on the held-out ClinVar split. Overall AlphaMissense and MutaPLM AUC are defined only for missense substitutions.

| Mutation type | MutaPLM | AlphaMissense | M2T Text | M2T Emb | Emb–Text $\Delta$ |
|---|---|---|---|---|---|
| All variants | 0.520* | 0.856* | 0.880 | 0.942 | +0.062 |
| Substitution | 0.520 | 0.856 | 0.869 | 0.910 | +0.041 |
| Deletion | – | – | 0.875 | 0.850 | −0.025 |
| Duplication | – | – | 0.818 | 0.970 | +0.152 |
| Frameshift | – | – | 0.734 | 0.879 | +0.145 |
| Insertion | – | – | 0.846 | 0.902 | +0.056 |
| Other | – | – | 0.856 | 0.914 | +0.058 |

Table 7: AUC by protein length (amino acids) on the held-out ClinVar split.

| Length (aa) | AlphaMissense | M2T Text | M2T Emb | Emb–Text $\Delta$ |
|---|---|---|---|---|
| 0–299 | 0.885 | 0.883 | 0.901 | +0.018 |
| 300–599 | 0.894 | 0.869 | 0.937 | +0.068 |
| 600–899 | 0.881 | 0.900 | 0.933 | +0.033 |
| 900–1199 | 0.820 | 0.889 | 0.963 | +0.074 |
| 1200–1499 | 0.775 | 0.895 | 0.976 | +0.080 |
| 1500–1799 | 0.826 | 0.871 | 0.935 | +0.065 |
| 1800–2099 | 0.745 | 0.843 | 0.932 | +0.089 |

formance, the text-based predictions yielded a significantly lower AUC of 0.85. This suggests that while the model encodes highly accurate information in its latent space, a substantial portion of this signal is lost during text generation. This finding suggests that future work should focus on better calibrating the alignment between latent representations and text, potentially through methods like reinforcement learning, to ensure a model's explanations are as accurate as its internal knowledge.

## 7 LATENT VS. TEXT PATHOGENICITY ("LOST IN TRANSLATION")

This section will be expanded with additional analysis in the camera-ready version. The key results and conclusions are already included below."

In the previous section, we observed that a simple classifier trained on Mutation2Text's latent protein embeddings achieves substantially higher pathogenicity AUC than a classifier based on the generated text. We now analyze this "lost-in-translation" gap in more detail. Overall latent vs. text performance. On the held-out ClinVar test set, the embedding-based classifier attains an AUC of 0.94, whereas the text-based classifier reaches 0.88. To understand which variants are most affected, we stratify results by mutation type and by protein length (Tables 6–7).

**By mutation type.**  Table 6 reports AUC for MutaPLM, AlphaMissense, and Mutation2Text's text and embedding classifiers across variant classes. The gap between embeddings and text is modest for single substitutions (0.910 vs. 0.869), but grows substantially for duplications and frameshifts (+0.152 and +0.145 AUC, respectively), where wild-type and mutant sequences diverge more dramatically. For deletions, the text classifier slightly outperforms the embedding classifier, suggesting that textual priors can sometimes compensate for noisier latent signals.

**By protein length.**  We also stratify performance by wild-type protein length in 300-residue bins (Table 7). As sequence length increases, the embedding-based AUC steadily improves from $\approx 0.90$ to $\approx 0.97$, indicating that the Perceiver resampler successfully leverages the additional context. In contrast, the text-based classifier remains nearly flat around 0.88–0.90 AUC. This pattern suggests that the latent space captures increasingly fine-grained pathogenicity signals as more sequence context becomes available, but the text head saturates at a lower performance ceiling.

**Implications.** Taken together, these results suggest a resolution or compression bottleneck: the Perceiver-based latent space captures rich pathogenicity information, especially for complex variants in long, well-labeled proteins, but translating this high-dimensional representation into a short natural- language explanation loses some of that signal. In practice, one can expose the embedding-based score directly as a calibrated pathogenicity estimate, while using the generated text as a complementary explanation layer. Closing this gap, e.g., via consistency losses between latent logits and text-implied predictions or RL-style fine-tuning with feedback on factuality remains an important direction for future work.

## 8 CONCLUSION

We introduced **Mutation2Text**, a multi-modal generative model that contrasts wild-type and mutant PLM embeddings to produce natural language answers to questions regarding the functional effect of mutations. Architecturally, a gated cross-attention block combines wild-type and mutant representations, a Perceiver Resampler enables length-invariant encoding of full proteins without truncation, and an LLM-aligned projector fuses protein features with natural language queries. To support instruction and evaluation at scale, we curated **Mutation2TextQA**, a large mutation-focused QA corpus mined from the literature, as well as ClinVarQA, a corpus of templated QA for pathogenic/benign coding variants.

Across benchmarks, Mutation2Text consistently outperforms strong baselines on *mutation effect* including pathogenicity and disease association prediction, mutation description, and open-ended QA, while remaining competitive on zero-shot descriptions of wild-type *protein function* from sequence alone. Beyond aggregate metrics, qualitative analyses show the model can articulate directional effects (e.g., loss- vs. gain-of-function) and link them to functional context such as domains, motifs, and interactions, moving variant interpretation from opaque scores toward transparent, rationale-driven explanations.

We release code, models, and datasets to facilitate reproducibility and to encourage community benchmarks on clinically relevant settings, with the long-term goal of assisting experts in turning raw sequence changes into actionable explanations.

## 9 ACKNOWLEDGMENTS

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

## A APPENDIX

### A.1 RELATED WORKS

**Protein Language Models (PLMs).** Protein language models (PLMs) such as ESM (Lin et al., 2022b), ProGen (Madani et al., 2020), ProtTrans (Elnaggar et al., 2021), and ProLLaMA (Lv et al., 2024) have demonstrated strong capabilities in learning sequence-level representations from large

protein databases (uni, 2021; Berman et al., 2000; Mistry et al., 2021). Trained on millions of wild-type sequences, these models capture structural motifs, domain-level organization, and evolutionary constraints. PLMs have enabled success in downstream tasks including stability prediction, contact map inference, and secondary structure classification. However, they are fundamentally mutation-agnostic: they are not trained to model the effect of mutations, and their predictions often rely on heuristic comparisons such as masked token log-likelihoods or pseudo-likelihood scores (Meier et al., 2021; Hou et al., 2025), which fail to reflect biological function or interpretability.

**LLM-Augmented Protein Reasoning.** Several recent works have explored integrating large language models (LLMs) with protein sequence understanding. BioMedGPT (Luo et al., 2023), Protein2Text (Jararweh et al., 2025), and Mol-Instructions (Fang et al., 2024) leverage instruction tuning and prompt engineering to map protein sequences to textual outputs. While these approaches demonstrate early signs of multimodal fluency, they are not designed to reason over mutations and do not model differences between wild-type and mutated proteins.

**Mutation-Aware Language Models.** Recent efforts have explored equipping language models with mutation reasoning capabilities. **MutaPLM** (Luo et al., 2024) fine-tunes protein language models using curated textual descriptions of single amino acid variants (SNVs) from UniProt. While effective for mutation annotation tasks, MutaPLM is restricted to SNVs, limiting its ability to model diverse mutation types such as insertions, deletions, or frameshifts. Its mutation reasoning is localized and not designed for contextual question answering or open-ended biological interpretation.

In contrast, **BioReason** (Fallahpour et al., 2025) operates at the DNA level, leveraging reference and variant sequences alongside textual queries to perform disease-centric sequence-to-sequence reasoning. It supports both SNVs and non-SNV variants (e.g., short indels), but its reasoning is primarily conditioned on gene-level context.

## A.2 PROTEIN SEQUENCE ENCODING

We employ the `esm3_sm_open_v1` variant of ESM-3 (Hayes et al., 2025) to encode wildtype and mutated protein sequences. Given a protein sequence $S = \{s_1, s_2, \ldots, s_L\}$ of length $L$, where $s_i$ represents an amino acid, ESM-3 outputs embeddings of dimension 1536 while being finetuned:

$$\mathbf{H}_{\text{wt}} = \text{ESM3}(S_{\text{wt}}) \in \mathbb{R}^{L \times 1536}, \quad \mathbf{H}_{\text{mut}} = \text{ESM3}(S_{\text{mut}}) \in \mathbb{R}^{L \times 1536}$$

where:

- $S_{\text{wt}}$ is the wild-type protein sequence.
- $S_{\text{mut}}$ is the mutated protein sequence.
- $s_i$ denotes the $i$-th amino acid in the sequence.
- $L$ is the length of the protein sequence.
- $\mathbf{H}_{\text{wt}} \in \mathbb{R}^{L \times 1536}$ is the sequence embedding matrix for the wild-type, with one 1536-dimensional embedding per residue.
- $\mathbf{H}_{\text{mut}} \in \mathbb{R}^{L \times 1536}$ is the sequence embedding matrix for the mutant.

## A.3 ATTENTION BLOCK TRAINING MODE

It is designed to operate under three training/inference modes, determined by the available protein sequence type in the input query (see Algorithm 1 for pseudocode):

- **Wild-type Only Mode** (`wt_only`): Used when only the wild-type sequence is present in the input query. In this case, the model applies *self-attention* solely to the wild-type sequence. No cross-attention or gating is applied, and only the wild-type representation is returned.

- **Mutation-only/ Delta Mode** (`delta_only`): Applied when only the mutated sequence is present in the query. Although the wild-type sequence is not explicitly mentioned in the input, it is still required internally to provide a reference for mutation. The model performs

*self-attention* on the mutated sequences and *cross-attention* from mutated to wild-type tokens as shown in the Attention Block of Figure 1. A learned *gating mechanism* integrates both signals into a delta representation, which is the only output used for downstream computation.

- **Full Mode** (`full`): Used when both wild-type and mutated sequences are provided. The model computes *self-attention* on each sequence independently and applies *cross-attention* from mutated to wild-type tokens. A *gating mechanism* fuses these outputs into a delta representation as it did in delta mode. This mode returns both the delta representation and the wild-type self-attention output.

### A.3.1 SHARED SELF-ATTENTION LAYER

$$\text{SelfAttn}(\mathbf{H}) = \text{MultiHead}(\mathbf{H}\mathbf{W}_Q, \mathbf{H}\mathbf{W}_K, \mathbf{H}\mathbf{W}_V)$$

where $\mathbf{W}_Q, \mathbf{W}_K, \mathbf{W}_V \in \mathbb{R}^{d \times d_k}$ are learned projection matrices shared across all sequences.

For any input sequence (either wild-type or mutated), the self-attention output is computed as:

$$\mathbf{O}_{\text{self}} = \text{SelfAttn}(\text{LayerNorm}(\mathbf{H}))$$

### A.3.2 CROSS-ATTENTION LAYER

$$\mathbf{Q}_{\text{cross}} = \text{LayerNorm}(\mathbf{H}_{\text{mut}})\mathbf{W}_Q^{\text{cross}} \tag{1}$$

$$\mathbf{K}_{\text{cross}}, \mathbf{V}_{\text{cross}} = \text{LayerNorm}(\mathbf{H}_{\text{wt}})\left[\mathbf{W}_K^{\text{cross}} \,\|\, \mathbf{W}_V^{\text{cross}}\right] \tag{2}$$

$$\mathbf{O}_{\text{cross}} = \text{softmax}\left(\frac{\mathbf{Q}_{\text{cross}}\mathbf{K}_{\text{cross}}^{\top}}{\sqrt{d_k}}\right)\mathbf{V}_{\text{cross}}$$

### A.3.3 ATTENTION-BASED GATING.

In modes where both $\mathbf{O}_{\text{self}}$ and $\mathbf{O}_{\text{cross}}$ are available (i.e., `full` and `delta_only`), a learned gating mechanism adaptively fuses both attention streams:

$$\mathbf{g} = \text{MultiHeadAttn}(\text{LayerNorm}(\mathbf{O}_{\text{self}}), \text{LayerNorm}(\mathbf{O}_{\text{cross}}))$$

$$\boldsymbol{\alpha} = \text{softmax}(\text{Linear}(\mathbf{g})) \in \mathbb{R}^2$$

$$\boldsymbol{\delta} = \alpha_1 \cdot \mathbf{O}_{\text{self}} + \alpha_2 \cdot \mathbf{O}_{\text{cross}}$$

**Final Output.** The fused output $\boldsymbol{\delta}$ is passed through a final linear projection and a gated residual connection:

$$\boldsymbol{\delta} \leftarrow \text{Linear}_{\text{out}}(\boldsymbol{\delta}) + \gamma \cdot \text{FFN}(\boldsymbol{\delta})$$

where $\gamma$ is a learned scalar gate controlling the residual flow.

## B PERCEIVER RESAMPLER

Our Perceiver Resampler operates on a set of $N = 16$ learnable latent queries that iteratively attend to the protein features through a lightweight stack of transformer-style cross-attention layers. This mechanism compresses high-dimensional, token-level representations into a compact set of semantically rich embeddings.

We apply the resampler independently to both the mutation-aware delta representation ($\boldsymbol{\delta}$) and the wild-type representation ($\mathbf{W}_{\text{out}}$), when available. Each produces a fixed-size output:

$$\mathbf{P}_{\text{protein}}^{\text{mut}}, \; \mathbf{P}_{\text{protein}}^{\text{wt}} \in \mathbb{R}^{16 \times 1536}$$

where:

---

**Algorithm 1** Attention Block Forward Pass

---

**Require:** $\mathbf{H}_{\text{mut}}, \mathbf{H}_{\text{wt}} \in \mathbb{R}^{L \times 1536}$
**Require:** mode $\in \{\texttt{full}, \texttt{delta\_only}, \texttt{wt\_only}\}$
**Ensure:** $\boldsymbol{\delta}, \mathbf{W}_{\text{out}}$
1: $\boldsymbol{\delta} \leftarrow$ None, $\quad \mathbf{W}_{\text{out}} \leftarrow$ None
2: **if** mode $\in \{\texttt{full}, \texttt{wt\_only}\}$ **then**
3: $\quad \mathbf{H}_{\text{wt}}^{\text{norm}} \leftarrow \text{LayerNorm}(\mathbf{H}_{\text{wt}})$
4: $\quad \mathbf{W}_{\text{out}} \leftarrow \text{SelfAttn}(\mathbf{H}_{\text{wt}}^{\text{norm}})$
5: $\quad \mathbf{W}_{\text{out}} \leftarrow \text{Linear}_{\text{out}}(\mathbf{W}_{\text{out}})$
6: **end if**
7: **if** mode $= \texttt{wt\_only}$ **then**
8: $\quad$ **return** None, $\mathbf{W}_{\text{out}}$
9: **end if**
10: **if** mode $\in \{\texttt{full}, \texttt{delta\_only}\}$ **then**
11: $\quad \mathbf{H}_{\text{mut}}^{\text{norm}} \leftarrow \text{LayerNorm}(\mathbf{H}_{\text{mut}})$
12: $\quad \mathbf{Q}_{\text{cross}} \leftarrow \mathbf{H}_{\text{mut}}^{\text{norm}} \mathbf{W}_Q^{\text{cross}}$
13: $\quad \mathbf{K}_{\text{cross}}, \mathbf{V}_{\text{cross}} \leftarrow \text{LayerNorm}(\mathbf{H}_{\text{wt}})[\mathbf{W}_K^{\text{cross}} \| \mathbf{W}_V^{\text{cross}}]$
14: $\quad \mathbf{O}_{\text{cross}} \leftarrow \text{MultiHeadAttn}(\mathbf{Q}_{\text{cross}}, \mathbf{K}_{\text{cross}}, \mathbf{V}_{\text{cross}})$
15: $\quad \mathbf{O}_{\text{self}} \leftarrow \text{SelfAttn}(\mathbf{H}_{\text{mut}}^{\text{norm}})$
16: $\quad \mathbf{g} \leftarrow \text{MultiHeadAttn}(\text{LayerNorm}(\mathbf{O}_{\text{self}}),$
$\qquad \text{LayerNorm}(\mathbf{O}_{\text{cross}}))$
17: $\quad \boldsymbol{\alpha} \leftarrow \text{softmax}(\text{Linear}(\mathbf{g}))$
18: $\quad \mathbf{mixed} \leftarrow \alpha_1 \cdot \mathbf{O}_{\text{self}} + \alpha_2 \cdot \mathbf{O}_{\text{cross}}$
19: $\quad \boldsymbol{\delta} \leftarrow \text{Linear}_{\text{out}}(\mathbf{mixed}) + \gamma \cdot \text{FFN}(\mathbf{mixed})$
20: **end if**
21: **return** $\boldsymbol{\delta}, \mathbf{W}_{\text{out}}$

---

- $N = 16$ is the number of learnable latent queries used by the Perceiver Resampler.

- $\boldsymbol{\delta}$ is the mutation-aware delta representation that captures the difference between wild-type and mutant protein embeddings.

- $\mathbf{W}_{\text{out}}$ is the output representation of the wild-type protein sequence.

- $\mathbf{P}_{\text{protein}}^{\text{mut}} \in \mathbb{R}^{16 \times 1536}$ is the resampled embedding set for the mutant protein, with 16 latent tokens each of dimension 1536.

- $\mathbf{P}_{\text{protein}}^{\text{wt}} \in \mathbb{R}^{16 \times 1536}$ is the resampled embedding set for the wild-type protein, structured in the same way.

## C MLP PROJECTOR

To bring the resampled protein embeddings into alignment with the LLM's input space, we employ a two-layer feedforward network. This MLP Projector transforms each of the 16 latent vectors from their original 1536 dimensions to the 4096-dimensional representation expected by the LLM:

$$\mathbf{H}_{\text{hidden}} = \text{GELU}(\mathbf{P}_{\text{protein}} \mathbf{W}_1 + \mathbf{b}_1), \quad \mathbf{E}_{\text{protein}} = \mathbf{H}_{\text{hidden}} \mathbf{W}_2 + \mathbf{b}_2$$

$$\mathbf{W}_1 \in \mathbb{R}^{1536 \times 3072}, \quad \mathbf{W}_2 \in \mathbb{R}^{3072 \times 4096}$$

where:

- $\mathbf{P}_{\text{protein}} \in \mathbb{R}^{16 \times 1536}$ is the output of the Perceiver Resampler, containing 16 latent vectors of dimension 1536.

- $\mathbf{W}_1 \in \mathbb{R}^{1536 \times 3072}$ and $\mathbf{b}_1 \in \mathbb{R}^{3072}$ are the parameters of the first linear layer, expanding the representation.

- $\text{GELU}(\cdot)$ is the Gaussian Error Linear Unit activation function.

---

**Algorithm 2** Multimodal Input Preparation with Mutation Feature Token

---

**Require:** Query, wildtype context, $\mathbf{E}_{\text{protein\_wt}} \in \mathbb{R}^{B \times 16 \times 4096}$, $\mathbf{E}_{\text{protein\_mut}} \in \mathbb{R}^{B \times 16 \times 4096}$, $\mathbf{E}_{\text{mut\_feat}} \in \mathbb{R}^{B \times 1 \times 4096}$

**Ensure:** $\mathbf{E}_{\text{input}} \in \mathbb{R}^{B \times T' \times 4096}$
 1: **for** each sample $b$ in batch **do**
 2:     Tokenize Query and Wildtype context: $\mathbf{x}_b = [x_1, \ldots, x_T]$
 3:     Map to embeddings: $\texttt{TokenEmbed}(\mathbf{x}_b)$
 4:     Identify token pairs: $(s_{wt}, e_{wt})$, $(s_{mut}, e_{mut})$, and feature position $p_{\text{feat}}$
 5:     Create insertion list: $\mathcal{I} \leftarrow \{(\text{wt}, s_{wt}, e_{wt}), (\text{mut}, s_{mut}, e_{mut}), (\text{feat}, p_{\text{feat}}, p_{\text{feat}})\}$
 6:     Sort $\mathcal{I}$ by start index
 7:     Initialize $\texttt{parts} \leftarrow [], \texttt{last\_idx} \leftarrow 0$
 8:     **for** each $(\text{type}, \text{start}, \text{end})$ in $\mathcal{I}$ **do**
 9:         Append $\texttt{TokenEmbed}(\mathbf{x}_b[\texttt{last\_idx} : \texttt{start} + 1])$ to parts
10:         **if** type $=$ wt **then**
11:           Append $\mathbf{E}_{\text{protein\_wt}}[b]$ {16 wild-type tokens}
12:         **else if** type $=$ mut **then**
13:           Append $\mathbf{E}_{\text{protein\_mut}}[b]$ {16 mutant tokens}
14:         **else if** type $=$ feat **then**
15:           Append $\mathbf{E}_{\text{mut\_feat}}[b]$ {1 mutation feature token}
16:         **end if**
17:         $\texttt{last\_idx} \leftarrow \text{end}$
18:     **end for**
19:     Append $\texttt{TokenEmbed}(\mathbf{x}_b[\texttt{last\_idx} :])$ to parts
20:     Concatenate all parts to form $\mathbf{E}_{\text{input}}[b]$
21: **end for**
22: **return** $\mathbf{E}_{\text{input}}$ (padded to maximum sequence length $T'$ across the batch)

---

- $\mathbf{H}_{\text{hidden}} \in \mathbb{R}^{16 \times 3072}$ is the intermediate hidden representation after the first projection and nonlinearity.

- $\mathbf{W}_2 \in \mathbb{R}^{3072 \times 4096}$ and $\mathbf{b}_2 \in \mathbb{R}^{4096}$ are the parameters of the second linear layer, mapping into the LLM's embedding space.

- $\mathbf{E}_{\text{protein}} \in \mathbb{R}^{16 \times 4096}$ is the final projected protein embedding, aligned with the LLM's token embedding dimension and ready for integration into the language model input stream.

## D ADDITIONAL MUTATION FEATURES

**ESM-based Difference Features.** We compute element-wise differences between ESM-3 embeddings of wild-type and mutant sequences to capture changes. For indels where the aligned wild-type and mutant sequences have different lengths, we first perform a sequence alignment and introduce a learned gap embedding $g \in \mathbb{R}^{1536}$ whenever one side has no corresponding residue. Let $e_i^{\text{wt}}$ and $e_i^{\text{mut}}$ denote the aligned ESM embeddings at position $i$; if there is a gap on one side, we set the missing embedding to $g$ and define the delta as

$$d_i = e_i^{\text{mut}} - e_i^{\text{wt}}.$$

The gap embedding $g$ is trained jointly with the rest of the model, allowing the delta features to represent both substitutions and indel structure in a unified way. Mutation metadata encoding. In addition to the continuous delta features, we encode categorical metadata including mutation type (substitution, insertion, deletion, frameshift), the normalized position of the mutation along the sequence, and the change in length (in residues). Mutation type is represented as a learned embedding of a one-hot vector, while position and size are encoded as scaled scalar features.

**Feature aggregation.** The per-position delta features are processed with a Perceiver resampler and pooled into a single latent token, which is concatenated with the metadata vector and projected by a two-layer MLP to the final 4096-dimensional mutation feature vector that is supplied to LLaMA-3 as a dedicated mutation token.

Table 8: **Prompt template for Mutation2TextQA question–answer generation.** We prompt DeepSeek-Chat to extract mutation-specific Q&A pairs from PubMed articles.

**[System prompt]** You are an expert biomedical assistant. You are analyzing scientific articles to help extract insights about protein mutations. For each protein mutation listed, first verify if it is mentioned in the article. Then, generate up to 10 insightful Q&A pairs. Questions should use placeholders `<mutation>` and `<protein>` and be biologically curious. Answers must be directly based on the article, not generic knowledge.
Each output should follow the json structure:

```
{
  "pubmed_id": "40949929",
  "new_mutations": [
    "NM_XXXXXX.X(GENE):c.XXX>X (p.A123V)"
  ],
"mutations": [
    {
  "protein": "<protein>",
  "mutation": "<mutation>",
  "discussed": "Yes" | "Maybe" | "No",
  "qa_pairs": [

    {
    "question": "What effect does
                 <mutation> have on the
                 structure of <protein>?",

  "answer": "It disrupts the
            ATP-binding pocket,
            reducing catalytic efficiency."
      }
    ]
    }
  ]
}
```

**[User prompt]**
```
document:  <article text>
mutations:  <list of mutation strings>
```

## E   HOMOLOGY-BASED TRAIN–TEST SPLIT

To avoid data leakage due to high sequence similarity, we performed a homology-aware train–test split. Specifically:

1. We grouped records by wild-type sequence.

2. Using pairwise global alignment from Biopython (Cock et al., 2009), we computed identity scores between each unique sequence pair.

3. We constructed a similarity graph.

4. We clustered the graph using connected components and randomly assigned 80% of clusters to training and 20% to testing.

Based on the identity score to training set, each test example was labeled into one of three difficulty tiers:

- **Easy**: $\geq 70\%$ identity

- **Medium**: 40–69% identity

- **Hard**: $< 40\%$ identity

## F  MUTATION2TEXTQA QUALITY CONTROL

Table 9 summarizes the effect of our LLM-based filter on Mutation2TextQA. Approximately 3.3% of initial QA pairs are removed because they are not directly supported by the source article, not answerable from the article alone, or mention proteins or mutations absent from the article.

Table 9: Mutation2TextQA quality-control summary.

| Metric | Value |
|---|---|
| Total QA pairs | 854,152 |
| Valid QA pairs | 826,065 (96.71%) |
| Invalid QA pairs | 28,087 (3.29%) |

Table 10 provides a taxonomy of hallucination types among the filtered pairs, illustrating that most rejected QAs arise from misinterpretation or contradiction of the source text rather than exotic failure modes.

Table 10: Taxonomy of hallucinations among filtered Mutation2TextQA pairs.

| Type | Percentage (%) |
|---|---|
| Misinterpretation | 48.38 |
| Contradiction | 32.78 |
| Exaggeration | 15.46 |
| External knowledge | 2.47 |
| Unspecified | 0.88 |
| Speculative | 0.03 |

## G  TRIPARTITE FACTUALITY EVALUATION ON MUTADESCRIBE

Table 11 reports the results of our tripartite evaluation protocol on the MutaDescribe test set, comparing Mutation2Text to MutaPLM. Human labels indicate the proportion of explanations judged Correct or Partially Correct by a blinded expert. LLM-judge scores are averaged over two independent models on a 0–5 scale, and CIDEr quantifies information content with respect to reference descriptions.

Table 11: Factuality evaluation on MutaDescribe.

| Model | Human (Correct/Partial) | LLM judge (0–5) | CIDEr |
|---|---|---|---|
| MutaPLM | 1% (0% / 1%) | 1.8 | 0.013 |
| Mutation2Text | 38% (10% / 28%) | 2.58 | 0.499 |

To assess the reliability of the LLM judges, we compute quadratic-weighted Cohen's $\kappa$ and Spearman correlation $\rho$ between their scores (Table 12). We observe strong agreement and significant correlation with human labels, supporting the use of LLM-as-judge as a scalable proxy for expert evaluation in our setting.

## H  BASELINES

We benchmark **Mutation2Text** against a diverse set of baselines, including *general-purpose protein LLMs*, *instruction-tuned multimodal LLMs*, *large foundation models*, and *mutation-aware architectures*. For the MutaDescribe benchmark, we strictly follow the original protocol (Luo et al., 2024) by training on the released training data and evaluating on the official test set. Our results are integrated into the original benchmark table for direct comparison (Table 1).

Table 12: Inter-judge reliability for LLM-based factuality scoring.

| Metric | Value | Interpretation |
|--------|-------|----------------|
| Quadratic $\kappa$ | 0.73 | Strong agreement |
| Spearman $\rho$ | 0.63 | Significant correlation |

In addition, we introduce **Mutation2TextQA**, a new benchmark specifically designed to evaluate reasoning over free-form mutation question–answer pairs. To assess the generalization of existing models, we evaluate some of the baselines on Mutation2TextQA, as well as on disease prediction and pathogenicity prediction tasks (Figure 3).

**General protein LLMs:** *ProLLaMA* and *Fine-tuned ESM-2* capture general protein sequence-level representations but lack explicit mechanisms for modeling mutational changes.

**Instruction-tuned multimodal LLMs:** Models such as *Mol-Instructions*, *Protein2Text*, and *BioMedGPT* apply instruction tuning for biological tasks such as protein function prediction and drug property predictions. While capable of generating fluent outputs, they are not trained for reasoning between wild-type and mutant proteins.

**Large foundation models.** *Galactica-6.7B* and *GPT-4* variants leverage large-scale pretraining and few-shot prompting. When augmented with external encoders like *ESM-2* or *OntoProtein* (Luo et al., 2024), GPT-4 demonstrates improved factual grounding but remains mutation-agnostic.

**Mutation-aware models.** *MutaPLM*, trained on UniProt mutation descriptions, is the strongest mutation-specific baseline. However, it is limited to single amino acid substitutions (SNV).

## I   DATASET STATISTICS

### I.1   MUTATION2TEXTQA DATASET STATISTICS

| Basic Metrics | |
|---------------|---|
| Total Records | 452,192 |
| Unique Mutations | 51,692 |
| Unique Genes | 4,004 |
| Unique PubMed IDs | 38,259 |
| Avg. QA Pairs / Mutation | 8.75 |

| Protein Sequences | |
|-------------------|---|
| Average Length | 612.1 amino acids |
| Minimum Length | 84 amino acids |
| Maximum Length | 1,972 amino acids |
| Standard Deviation | 398.0 amino acids |

| Conversations & Responses | |
|---------------------------|---|
| Conversation Length | 2 turns (fixed) |
| Average Response Length | 22.4 words |
| Min / Max Response Length | 4 / 55 words |
| Response Std Dev | 5.4 words |

| Question Types | |
|----------------|---|
| Other | 78.7% |
| Affect | 16.6% |
| Functional Impact | 1.4% |
| Evidence | 1.4% |
| Location | 0.9% |
| Stability | 0.6% |
| Import | 0.2% |

## I.2 ALL TRAINING DATASET STATISTICS

Table 14: Aggregated statistics across Mutation2TextQA, MutaDescribe, and ClinVarQA.

| Basic Metrics | |
|---|---|
| Total Records | 1,211,584 |
| Unique Mutations | 336,570 |
| Unique Genes | 13,656 |
| Unique PubMed IDs | 38,034 |
| Avg. QA Pairs / Mutation | 3.86 |
| **Dataset Source Distribution** | |
| ClinVar | 594,156 (49.0%) |
| MutaDescribe | 165,236 (13.6%) |
| Mutation2TextQA | 452,192 (37.4%) |
| **Protein Sequences** | |
| Average Length | 753.8 amino acids |
| Minimum Length | 10 amino acids |
| Maximum Length | 2,000 amino acids |
| Standard Deviation | 454.9 amino acids |
| **Conversations & Responses** | |
| Conversation Length | 2 turns (fixed) |
| Average Response Length | 15.3 words |
| Min / Max Response Length | 2 / 332 words |
| Response Std Dev | 13.9 words |

## J PROTEIN2TEXT-QA BASELINES

We compare our model to different baselines throughout the manuscript. We mainly focus on two types of baselines: general-purpose LLMs and protein-specific LLMs. The general-purpose LLMs were used as a measure of data leakage, identifying the amount of information leaked from the prompt into the generated answer. Second, we assess protein-specific LLMs that use protein sequences and a text prompt as input. We now provide a high overview of the baselines and the prompting mechanism.

**GPT4o-mini** OpenAI et al. (2023). The model is a variant of the GPT4 family with a reduced number of parameters. We used the OpenAI API to generate responses in this manuscript where we feed the prompt and the sequence as input. We set the role to "*You are an expert assistant for protein-related inquiries*". The average response time is 30 seconds per query. We launched multiple processes per day for multiple days until the maximum number of tokens quota was reached.

**LLaMA3.1-8B-Instruct** (Dubey et al., 2024). LLaMA3.1-8B-Instruct [1] is a general multilingual model trained using instruction tuning to perform reasoning tasks. We utilize the same prompt structure used to query GPT4o-mini to extract responses from the model. We use the released model checkpoints from HuggingFace to extract responses. The average request time is 30 seconds per prompt on an 80GB H100.

**BioMedGPT** Luo et al. (2023). BioMedGPT is a multimodal LLM that integrates molecular structures, protein sequences, and natural language text. The model aligns the three modalities to perform cross-modal tasks about proteins and molecular compounds. The model utilizes LLaMA2 Touvron et al. (2023) as the LLM base model. The training data was extracted from different sources such as PubMed Central (PMC), PubChem Kim et al. (2022), and UniProt Consortium (2022). We utilize the weights and default parameters released by the authors to perform inferencing. The inference time is 0.09 seconds per query on an 80GB H100.

**Mol-Instruction** Fang et al. (2024). Similarly, Mol-Instruction is a multimodal LLM that integrates text, molecular compounds, and protein sequences. The model utilizes GPT3.5 to generate a QA dataset about proteins and compounds from PubMed articles. We utilize the LoRA weights published by the authors and the LLaMA-2-7b-chat-hf model from HuggingFace to perform inferencing. We utilize the default parameters as found in the released evaluation script. The approximate inferencing time is 18.17 seconds per query on an 80GB A100.

---

[1] https://huggingface.co/blog/llama31?utm_source=chatgpt.com

**ProtT3** Liu et al. (2024). ProtT3 utilizes multimodal projection to align between protein amino acid sequences and natural language text. The model is trained in two stages: protein-text retrieval and protein-text generation. During the first stage, contrastive learning objectives are utilized to extract protein features that match the description. Then, the LLM model is trained using LoRA to perform generative tasks. The authors release three different checkpoints for different tasks. We utilize the checkpoint released by the author for the QA task. The response time is 0.14 seconds per query on an 80GB H100.

Table 15: Example question–answer pairs from the **ClinVar-QA** dataset. All answers are auto-filled from structured ClinVar metadata.

| Question | Answer |
|---|---|
| What condition is most strongly associated with `<mutation>`? | The associated condition is Niemann-Pick disease type C1. |
| What kind of mutation is `<mutation>`? | It is classified as a missense mutation. |
| Which phenotype is linked to `<mutation>` in `<protein>`? | Studies have connected this variant with hypertrophic cardiomyopathy. |
| Is `<mutation>` known to be associated with any clinical condition? | This mutation has shown association with Marfan syndrome in affected individuals. |
| Can you describe the variant type of `<mutation>`? | This is a deletion-type change at the DNA level. |

## K HYPERPARAMETERS

### K.1 PRETRAINING (ADAPTERS ONLY)

- **Base LLM**: Meta-Llama-3.1-8B-Instruct
- **Protein encoder**: ESM3 (esm3_sm_open_v1)
- **Adapters**: Gated Cross-Attention (on), Perceiver Resampler (on), Projector type: mlp2x_gelu
- **Adapter dims**: num_media_tokens=16, gca_output_dim=512, resampler_output_dim=1536, num_heads=8, perceiver_depth=6, dim_head=64, ff_mult=4
- **Training**: epochs=1, per_device_train_batch_size=180, gradient_accumulation_steps=1
- **Optimization**: AdamW, learning_rate=$2.0 \times 10^{-4}$, weight_decay=0.0, warmup_ratio=0.03, lr_scheduler=cosine
- **Precision/Dist.**: bf16 (on), tf32 (on), gradient_checkpointing (on), DeepSpeed ZeRO-2

### K.2 LORA FINETUNING

- **LoRA**: enable=true, r=16, $\alpha$=32, dropout=0.05, targets={q_proj, k_proj, v_proj, o_proj}
- **Adapters during LoRA**: tune_mm_mlp_adapter=true (GCA/Resampler/Projector trainable)
- **Training**: epochs=5, per_device_train_batch_size=58, gradient_accumulation_steps=1
- **Optimization**: AdamW, learning_rate=$1.0 \times 10^{-5}$, weight_decay=0.0, warmup_ratio=0.03, lr_scheduler=cosine
- **Precision/Dist.**: bf16 (on), tf32 (on), gradient_checkpointing (on), DeepSpeed ZeRO-2
- **Logging/CKPT**: logging_steps=100, save_steps=1000, save_total_limit=3

### K.3 PATHOGENICITY CLASSIFIER (ESM → GCA → RESAMPLER → MLP) (FOR LIMITATION EXPERIMENT)

- **Backbone features**: Resampler output flattened (num_media_tokens=16, token_dim=1536)
- **Classifier**: MLP with token resampler (attention pooling to 1 token), hidden sizes [1024, 256], dropout=0.1
- **Training (typical)**: batch_size=128–1024, epochs=200, patience=8
- **Optimization**: AdamW, learning_rate=$3 \times 10^{-4}$, weight_decay=0.01
- **Precision/Dist.**: optional bf16, optional DeepSpeed (2 GPUs)

Table 16: **Qualitative analysis.** Comparing responses of Mutation2Text with baseline models. Correct answers are in blue and incorrect ones in red.

| Question (ID) | Ground Truth | Model Outputs |
|---|---|---|
| **Q1:** Does `<mut_protein>` affect the same functional pathway as other known TPO mutations? | Yes, like other TPO mutations, this variant disrupts thyroid hormonogenesis, leading to congenital dyshormonogenetic hypothyroidism with goiter. | **Mutation2Text:** Yes, like other TPO mutations (e.g., p.Gly105Arg), it disrupts iodide organification and thyroid hormone synthesis.
ProLLama: MRALAVLSVTLVMA....
BioMedGPT: May cause a mild form of congenital dyserythropoietic anemia.
Mol-Instructions: "Yes, the mutation affects the same functional pathway as other known TPO mutations." |
| **Q2:** What is the inheritance pattern of `<mut_protein>` in `<wt_protein>`? | Autosomal recessive inheritance, confirmed by homozygous patient. | **Mutation2Text:** Autosomal recessive, requiring biallelic variants for disease manifestation.
ProLLama: SMAD/ERFDFSHPIIITSDAG...
BioMedGPT: Maternally inherited.
Mol-Instructions: "The inheritance pattern of the mutation in the wild-type protein is autosomal dominant." |
| **Q3:** What is the functional consequence of `<mut_protein>` on mitochondrial RNA import? | The mutation significantly decreases the import of 5S rRNA and MRP RNA into mitochondria, as shown by RNA hybridization experiments in subject fibroblasts. | **Mutation2Text:** $tRNA^{Lys}$ is not imported into mitochondria in cells with the p.R132C mutation, indicating impaired RNA import function.
Protein2Text: The mutation in the protein is associated with impaired mitochondrial RNA import.
BioMedGPT: Does not play a role in the assembly of the mitochondrial ribosome.
Mol-Instructions: The functional consequence of the mutation is a loss of function in the protein's ability to bind to its ligand. |

### K.4 TRAINABLE PARAMETER COUNTS

- **Adapters Pretraining**: 39.65M trainable parameters.
- **Finetuning**: 46.18M trainable parameters.

## L MUTATION2TEXTQA DATASET ANALYSIS

Unlike previous datasets, we extract free-form QA pairs that address protein mutations as mentioned in scientific literature. Here, we analyze the dataset diversity to show its lexical and semantic richness across different topics.

### L.1 EXPERIMENTAL SETUP.

To analyze the dataset diversity, we first randomly sampled 20,000 questions from the dataset for semantic analysis. Questions were then embedded using BioMedBERT (Gu et al., 2021), a model pretrained on biomedical literature. Token embeddings were averaged (mean pooling) to form question-level embeddings. K-means clustering (20 clusters) was applied to these embeddings, with dimensionality reduction via UMAP (McInnes et al., 2020) for visualization. Clusters were annotated by extracting the 15 most representative questions (nearest to each cluster centroid). The topics for the 20 clusters were found by feeding the list of representative questions to GPT-4. We experimented with the number of representative questions and performed manual inspection to choose the appropriate number.

### L.2 FINDINGS.

Figure 3a shows the distribution of questions across mutation-related topics extracted in an unsupervised manner. With cluster 15 being the highest and cluster 17 being the lowest, the plot provides an overview of topic prominence and reveals which aspects of mutation-related literature are more heavily represented.

Figure 3b displays the semantic landscape of question embeddings in the datasets. The plot displays clear spatial separation into over twenty biologically meaningful and distinct topics, reflecting both breadth and depth of coverage. Each point represents a question and its position in the UMAP space reflects semantic similarity based on its embeddings. Most of the clusters form dense and coherent regions such as clusters 0,

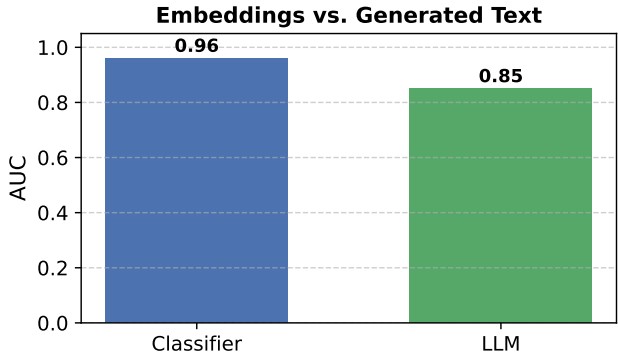

Figure 4: Performance is higher when training directly on embeddings from resampler (0.96) compared to keyword-based evaluation of generated text (0.85).

4, and 15, while others share some overlaps, indicating potential thematic proximity or shared vocabulary. For example, Clusters 0, 1, 3, and 10 highlight the cascade of mutation effects from molecular (protein activity) to cellular, tissue, and clinical outcomes, indicating the biological progression from molecular dysfunction to phenotypic expression and disease manifestation. Such semantic diversity supports downstream LLM training by facilitating models capable of reasoning across diverse biological functions and contexts.

### L.3 EXPERIMENTAL SETUP.

To analyze the dataset diversity, we first randomly sampled 20,000 questions from the dataset for semantic analysis. Questions were then embedded using BioMedBERT (Gu et al., 2021), a model pretrained on biomedical literature. Token embeddings were averaged (mean pooling) to form question-level embeddings. K-means clustering (20 clusters) was applied to these embeddings, with dimensionality reduction via UMAP (McInnes et al., 2020) for visualization. Clusters were annotated by extracting the 15 most representative questions (nearest to each cluster centroid). The topics for the 20 clusters were found by feeding the list of representative questions to GPT-4. We experimented with the number of representative questions and performed manual inspection to choose the appropriate number.

### L.4 FINDINGS.

Figure 3a shows the distribution of questions across mutation-related topics extracted in an unsupervised manner. With cluster 15 being the highest and cluster 17 being the lowest, the plot provides an overview of topic prominence and reveals which aspects of mutation-related literature are more heavily represented.

Figure 3b displays the semantic landscape of question embeddings in the datasets. The plot displays clear spatial separation into over twenty biologically meaningful and distinct topics, reflecting both breadth and depth of coverage. Each point represents a question and its position in the UMAP space reflects semantic similarity based on its embeddings. Most of the clusters form dense and coherent regions such as clusters 0, 4, and 15, while others share some overlaps, indicating potential thematic proximity or shared vocabulary. For example, Clusters 0, 1, 3, and 10 highlight the cascade of mutation effects from molecular (protein activity) to cellular, tissue, and clinical outcomes, indicating the biological progression from molecular dysfunction to phenotypic expression and disease manifestation. Such semantic diversity supports downstream LLM training by facilitating models capable of reasoning across diverse biological functions and contexts.

To further probe the distinctiveness of these clusters, we use type-token ratio (TTR) curves to evaluate the vocabulary richness within each topic. Figure 3c plots the dataset's type-token ratio (TTR). Lower curves in the plot indicate more repetition in the vocabulary as opposed to the top curves. This lexical analysis complements the semantic view by revealing how diverse or repetitive the language is within each cluster, thus providing insight into whether conceptually close clusters (e.g., 0 and 1 or 3 and 10) also share similar linguistic complexity or rely on specialized versus general vocabulary.

