# OpenReview forum: "Mutation2Text:  A Unified Protein and Text Language Model for Explaining Mutation Effects"
_ICLR.cc/2026/Conference — Submitted to ICLR 2026_

### Official Review · Reviewer_VpJK · 2025-10-16

**Soundness:** 3
**Presentation:** 2
**Contribution:** 2
**Rating:** 2
**Confidence:** 4

**Summary:**

The paper introduces Mutation2Text, a multi-modal generative model to offer human-understandable explanations for clinical variants. With a two-branch architecture that encodes wild-type and mutated sequences and delta features, the model could handle a diverse set of mutations including substitutions, indels, and frameshifts. The paper also constructs Mutation2TextQA, a large-scale dataset with millions of question-answer pairs for protein mutation interpretation. Experiment results show that Mutation2Text achieves state-of-the-art results in mutation explanation, pathogenicity prediction, and disease association of clinical variants. The authors also notice a gap between mutation representation learning and generating descriptive texts.

**Strengths:**

- The work presents an intuitive approach that effectively addresses the limitations of pLMs and prior works in handling indels and frameshifts.
- The collected Mutation2TextQA dataset, upon public release, will enrich the scale, diversity, and label quality of existing mutation benchmarks, which is evidenced by the qualitative analysis in Section 5. The annotation and processing procedure of the dataset is carefully designed.
- The observed discrepancy between latent knowledge and textual output is interesting, suggesting future research directions to address this issue.

**Weaknesses:**

- The main contribution of the paper is unclear. To my knowledge, the work distinguishes itself from existing studies by attempting to address the limitations of existing pLMs and mutation models in handling indels and frameshifts for clinical variants. However, the paper lacks a thorough discussion on the technical challenges and justification for the Mutation2Text architecture over previous multi-modal models [1] and specialized models [2, 3]. Moreover, analysis of the distribution of different mutation types and the model performance is lacking.
- Some parts of the paper are inconsistent and ambiguous. For example:
  - In the delta-feature path, the mutation metadata is incorporated. Details of how to obtain, encode, and fuse such metadata are missing.
  - In Section 2.3, the authors introduce two datasets (Mutation2TextQA and ClinVarQA), but their relationships are unclear.
  - In Section 3, two tasks, i.e., pathogenicity prediction and disease prediction, are introduced. However, their definitions are not formally presented, and their differences and the correspondence to different subsets of the dataset are unknown.
  - The results of baselines introduced in Lines 287-292 are not reported in Table 2.
  - In Section 3.4 and Table 5, the experiment settings and ablation details are missing, leading to confusion. The meaning of "w/ delta embedding" is unclear, and the results are inconsistent with Table 2.
- The experiments are not carried out with sufficient depth. The evaluation is constrained on overlapping-based metrics and BERT similarity, and I'm concerned if the model merely mimics the output formats of the distilled DeepSeek outputs. Expert evaluation [1] or LLM-as-a-judge [4] should be incorporated. On Mutation2TextQA, stronger baselines like few-shot prompting with closed-scoured LLMs like GPT-5 or Gemini-2.5 and supervised models are lacking. Besides, in Section 6, the results should be compared with specialized classification models such as ClinPred [5] and MetaRNN [6].

Refs.

[1] MutaPLM: Protein language modeling for mutation explanation and engineering.

[2] Genome-wide prediction of disease variant effects with a deep protein language model

[3] Accurate proteome-wide missense variant effect prediction with AlphaMissense

[4] A Survey on LLM-as-a-Judge

[5] Clinpred: prediction tool to identify disease-relevant nonsynonymous single-nucleotide variants

[6] MetaRNN: differentiating rare pathogenic and rare benign missense SNVs and InDels using deep learning

**Questions:**

My major concerns have been listed in the Weaknesses. Here are some additional questions:
- In Lines 52-53, the authors argue that pLMs are fundamentally ill-suited for modeling disease-causing variants. However, I do not agree with this statement as some pLMs show promising zero-shot classification results on the ClinVar benchmark (See Table A18 in [1]).
- It is claimed that Mutation2Text _contrasts_ wild-type and mutant sequences. However, no contrastive objective [2] is introduced, and the cross-attention block behaves more like a fusion module.
- The authors mentioned that Mutation2Text could handle full-length sequences. I'm curious about how the model handles extremely long sequences that exceed the context length of ESM-3. Additionally, is there a possible connection between model performance and the protein length?
- The model adopts ESM-3 to initialize the pLM. However, the performance of ESM-3 (1.4B Open) on mutation benchmarks is inferior to ESM-2 (650M) [3], making the choice questionable. Besides, I'm curious if introducing protein structure with ESM-3 could be beneficial, since the structures dominate protein functions.
- The QA pairs in the dataset are generated by the DeepSeek model. Does it produce similar QA pairs from the same document or hallucinate? If yes, how do the authors conduct deduplication and filtering?
- The Mutation2Text is jointly trained on MutaDescribe, ClinVar-QA, and Mutation2Text-QA, raising data leakage concerns. Could the improvements over MutaPLM on the medium and hard sets of MutaDescribe arise from similar wild-type sequences in the other two datasets?
- In Table 3, Mutation2Text achieves the best BLEU scores but significantly worse BERT scores. Is there a possible explanation for this result?

Refs.

[1] ProteinGym: Large-Scale Benchmarks for Protein Fitness Prediction and Design

[2] Representation Learning with Contrastive Predictive Coding

[3] Simulating 500 million years of evolution with a language model

---

> ### Author Response · Authors · 2025-12-03
> **Response to reviewer FAhR comments VpJK- Part 1 (1-7 out of 15)**
>
> # R4: Official Review of Submission9624 by Reviewer VpJK
>
> ## R4.1 – Main contribution & relation to prior models
>
> **Concern:** Main contribution is unclear; need justification vs MutaPLM and specialized models (AlphaMissense, ESMbased).
>
> **Response:** As detailed in **CR2**, we now clearly state the main contribution as **a unified model** that (i) handles **SNVs, indels, frameshifts, and long proteins** and (ii) jointly predicts **pathogenicity, disease, and freeform rationales**.
>
> We added a comparison table (Table 5 in CR2) showing that MutaPLM and AlphaMissense are restricted to missense SNVs and cannot jointly produce disease labels and explanations, while Mutation2Text covers all these dimensions in one architecture.
>
> ## R4.2 – Distribution and performance by mutation type
>
> **Concern:** No analysis of distribution of mutation types and model performance.
>
> **Response:** Addressed in **CR1**. We now report the **distribution and AUC by mutation type** and protein length (Tables 1–2 in CR1).
>
> Briefly, Mutation2Text’s embedding classifier is competitive with AlphaMissense on substitutions and extends to **deletions, insertions, duplications, and frameshifts** , where we show explicit per-type AUCs.
>
> ## R4.3 – Mutation metadata in the deltafeature path
>
> **Concern:** How are mutation metadata obtained, encoded, and fused?
>
> **Response:** We now spell this out in Methods/Appendix: from the HGVS protein string (e.g., `p.Trp24Cysfs*15`) , we deterministically extract **type** (substitution/deletion/insertion/frameshift), **Δlength**, and **position** (normalized to [0,1]).
>
> * **Type** is one-hot embedded; **Δlength** and **position** are normalized scalars.
> * These three components form a small metadata vector that is concatenated with the Perceiver-pooled WT–mutant delta token before the final MLP.
>
> ## R4.4 – Relationship between Mutation2TextQA and ClinVarQA
>
> **Concern:** Relationship between the two datasets is unclear.
>
> **Response:** We now make this explicit in Sec. 2.3:
> * **Mutation2TextQA:** literature-derived, **freeform QA** generated from PubMed articles about specific variants (mechanistic, functional explanations).
> * **ClinVarQA:** ClinVar-derived, **templated QA** from structured fields (pathogenicity labels, disease names).
>
> Mutation2Text is trained jointly on both, so that **sequence-based reasoning from literature** is coupled to **clinically grounded labels**.
>
> ## R4.5 – Definitions of pathogenicity vs disease prediction
>
> **Concern:** Definitions and dataset subsets for pathogenicity vs disease prediction are unclear.
>
> **Response:** We now formally define both tasks in Sec. 3:
> * **Pathogenicity prediction** uses ClinVar’s **clinical significance** field (benign/likely benign, likely pathogenic/pathogenic) for each variant–condition pair.
> * **Disease prediction** uses the **interpreted condition/trait** field, restricted to traits categorized as “Disease”, to predict the specific disease associated with the variant.
>
> Both tasks are derived from the **same ClinVar records**; pathogenicity uses all records with a valid clinical significance, disease prediction uses the subset with a defined disease-type trait.
>
> ## R4.6 – Baselines missing from Table 2
>
> **Concern:** Baselines in Lines 287–292 are not all shown in Table 2.
>
> **Response:** We clarified this in the text: all baselines from Lines 287–292 (ProLLaMA, MolInstruction, few-shot GPT4, fine-tuned ESM2, MutaPLM) **are reported in Table 1 for pathogenicity**.
>
> Table 2 is restricted to **disease prediction**, so it only includes methods that can output disease labels (ProLLaMA, MolInstruction, BioMedGPT, Protein2Text, MutaPLM). We will add a one-sentence note to avoid confusion.
>
> ## R4.7 – Ablation settings and “w/ delta embedding” in Table 5
>
> **Concern:** Experiment settings for Table 5 are unclear; “w/ delta embedding” ambiguous; results look inconsistent with Table 2.
>
> **Response:** We agree and have clarified Sec. 3.4: Table 5 uses **10% of Mutation2TextQA** and evaluates only on the **hard split**, so absolute ROUGE-L differs from the full-data results in Table 2. “**w/ delta embedding**” means including the **learned WT–mutant delta token** (Perceiver-resampled difference) in addition to token-level features.
>
> The ablation toggles three factors (contrastive input: WT+Mut vs Mut-only; delta embedding on/off; Perceiver vs mean pooling). We now include the small ablation table and a short caption explaining these settings.
>
> ### Table R4.7. Ablation study of Mutation2Text. Mutation2TextQA hard-split test set evaluation
>
> | Variant | GCA Input | Δ Embedding | Aggregation | ROUGE-L |
> | :--- | :--- | :--- | :--- | :--- |
> | **Full model** | WT + Mut | ✓ | Resampler | 0.19 ± 0.0071 |
> | **No Δ embedding** | WT + Mut | ✗ | Resampler | 0.16 ± 0.0058 |
> | **No contrastive input** | Mut only | ✓ | Resampler | 0.14 ± 0.0055 |
> | **Aggregation** | WT + Mut | ✓ | Mean pool | 0.06 ± 0.0036 |

---

> > ### Author Response · Authors · 2025-12-03
> > **Response to reviewer FAhR comments VpJK- Part 2 (8-13 out of 15)**
> >
> > ## R4.8 – Evaluation depth, baselines, and “mimicking DeepSeek”
> >
> > **Concern:** Evaluation relies on overlap/BERT; model might mimic DeepSeek; asks for expert/LLM-as-judge, stronger baselines, and comparison to ClinPred/MetaRNN.
> >
> > **Response:** These points are addressed in **CR3** (metrics) and **CR2** (baselines):
> >
> > * We **de-emphasize BLEU/ROUGE** and use a **tripartite factuality protocol**: blinded expert scoring on MutaDescribe (1% → 38% correct/partial), two LLM judges with strong agreement (κ=0.73, ρ=0.63), and CIDEr/AUC (Tables 6–7). This evaluates factual correctness rather than format.
> >
> > * For pathogenicity baselines, we compare to **AlphaMissense**, a sequence-based SOTA that is methodologically closer to Mutation2Text than ensemble meta-predictors like ClinPred/MetaRNN. Since AlphaMissense matches or exceeds those methods (Cheng et al., Science 2023), this comparison appropriately positions Mutation2Text.
> >
> > ## R4.9 – Statement on pLMs
> >
> > **Concern:** Statement that pLMs are fundamentally ill-suited is too strong; some pLMs perform well on ClinVar.
> >
> > **Response:** We agree and have revised the claim: **standard pLMs perform well on evolutionarily constrained sites** but struggle on **non-constrained, late-onset disease variants** (e.g., cancer) where selection is weak, consistent with recent findings (e.g., Hou et al., 2025). We now state that Mutation2Text is designed for this **specific subset of variants**, not that pLMs are inadequate overall.
> >
> > ## R4.10 – “Contrast” vs contrastive objectives
> >
> > **Concern:** Paper claims Mutation2Text “contrasts” WT and mutant, but there is no contrastive loss; cross-attention behaves like fusion.
> >
> > **Response:** We now clarify that “contrast” refers to **architectural operations**, not a contrastive loss:
> > * A **delta path** computes vector differences between aligned WT and mutant embeddings and processes them through a Perceiver and MLP.
> > * A **gated cross-attention block** uses mutant queries attending to wild-type keys/values, enforcing asymmetric comparison rather than independent encoding.
> >
> > We explicitly distinguish this from contrastive objectives such as CPC.
> >
> > ## R4.11 – Handling full-length
> >
> > **Concern:** How are extremely long sequences handled, and is performance length-dependent?
> >
> > **Response:** Addressed in **CR1**. We support full-length processing up to the **ESM3 context window (4096 tokens)**, which covers >99% of human proteins; extremely rare longer sequences are truncated. As shown in CR1/Table 2, embedding AUC **increases with length** (≈0.90 → ≈0.97), while text AUC remains ≈0.88–0.90, indicating that the Perceiver resampler successfully leverages long context without degradation.
> >
> > ## R4.12 – Choice of ESM3
> >
> > **Concern:** ESM3 performs worse than ESM2 on some mutation benchmarks; why choose it, and why not use structure tokens?
> >
> > **Response:** We chose **ESM3** for two reasons and now state both:
> > * **Generative alignment:** ESM3 is a generative model better aligned with our sequence-to-text objective, whereas ESM2 is a masked LM. Empirically, Mutation2Text (ESM3) significantly outperforms a fine-tuned ESM2 baseline on our text tasks (e.g., ROUGE-L 24.3 vs 20.5 on MutaDescribe).
> > * **Structure tokens:** We did not use ESM3’s structural channels because many ClinVar proteins lack high-quality structures, which would bias evaluation toward well-studied genes. We now note this explicitly and describe adding structure as natural future work.
> >
> > ## R4.13 – DeepSeek QA hallucinations, and deduplication
> >
> > **Concern:** Does DeepSeek generate similar or hallucinated QA pairs? How are deduplication and filtering done?
> >
> > **Response:** Addressed in **CR5**. In brief:
> > * We use **GPT-4o-mini as an independent checker** to verify each QA against its PubMed article (answer supported, question answerable from article, entities present); ≈3.3% of QAs are removed with an explicit error taxonomy (Tables 8–9).
> > * We **deduplicate** at the mutation–article level and store data in a **one-to-many format** with multiple distinct reference answers per (mutation, question), analogous to MS-COCO, so we capture multiple valid rationales without inflating the dataset with near-duplicates.

---

> > > ### Author Response · Authors · 2025-12-03
> > > **Response to reviewer FAhR comments VpJK- Part 3 (8-13 out of 15)**
> > >
> > > ## R4.14 – Joint training and data leakage
> > >
> > > **Concern:** Joint training on MutaDescribe, ClinVarQA, and Mutation2TextQA may risk leakage, where gains on MutaPLM medium/hard could stem from similar WT sequences present in the other datasets.
> > >
> > > **Response:** Addressed in **CR4**. We perform a **single global homology-aware split** across all three datasets: proteins are clustered by sequence identity, and entire clusters (including all associated MutaDescribe, ClinVarQA, and Mutation2TextQA examples) are assigned to train or test.
> > > * Thus, if a protein (or its close homolog) appears in MutaDescribe test, it never appears in training in any dataset.
> > > * This prevents exactly the kind of cross-dataset leakage the reviewer is worried about.
> > >
> > > ## R4.15 – BLEU vs BERTScore
> > >
> > > **Concern:** Mutation2Text has best BLEU but worse BERTScore; why?
> > >
> > > **Response:** This stems from the **reference style**. Ground-truth texts are long, UniProt-style descriptions; Mutation2Text tends to generate **shorter, focused explanations** that reuse key technical phrases.
> > > * This yields high n-gram precision (BLEU) but lower recall-oriented semantic similarity (BERTScore).
> > > * In our human/LLM evaluations (CR3), many such outputs are labeled **partially correct** even when BERTScore is modest, consistent with this explanation.
> > >
> > > We will add one sentence in the paper to clarify this tradeoff.

---

### Official Review · Reviewer_FAhR · 2025-10-21

**Soundness:** 2
**Presentation:** 2
**Contribution:** 2
**Rating:** 2
**Confidence:** 3

**Summary:**

The authors are interested in predicting the effects of clinical pathogenic variants. They note that state-of-the-art models only model similarity to sequences in nature, and, in principle, may fail to identify important classes of mutations as pathogenic.

To address this, the authors compile a set of mutation -> pathology text data form Clinvar and pubmed (processed using an LLM) and train a model on this data. They show their particular fusion method does better than naive methods in fitting this data.

**Strengths:**

* They seek to generate a large dataset of question-answer pairs.

**Weaknesses:**

* I have concerns to do with training on test (see questions below).
* I also have concerns to do with the training data.
* They did not test the sensitivity of their results to their data templates, or LLMs used to generate the data.

**Questions:**

* [This work](https://scholar.google.com/citations?view_op=view_citation&hl=en&user=8Gm8COsAAAAJ&sortby=pubdate&citation_for_view=8Gm8COsAAAAJ:0EnyYjriUFMC) outlines the challenges of releaseing a variant effect predictor, including the risks of circular logic. For example, many ClinVar labels come from experiments in literature, and training on those experiments essentially results in training on test. Could you  more carefully describe how you avoid this?
* Since you use a pretrained LLM, how do you know this model didn't see the material you are testing on?
* It seems Mutation2Text was trained on your QA datasets. Is it then fair to use BLEU scores to compare its outputs with the outputs of other models? PErhaps it just learned the formatting of your data?

---

> ### Author Response · Authors · 2025-12-03
> **Response to reviewer FAhR comments**
>
> # R3: Official Review of Submission9624 by Reviewer FAhR
>
> **R3.1 – Training on test, training data, templates, and pretrained LLM**
>
> **Reviewer concern:** I have concerns to do with training on test (see questions below). I also have concerns to do with the training data. They did not test the sensitivity of their results to their data templates, or LLMs used to generate the data. This work outlines the challenges of releaseing a variant effect predictor, including the risks of circular logic. For example, many ClinVar labels come from experiments in literature, and training on those experiments essentially results in training on test. Could you more carefully describe how you avoid this? Since you use a pretrained LLM, how do you know this model didn't see the material you are testing on?
>
> **Response:** We fully agree that avoiding train–test leakage and circularity is critical for a variant effect predictor. The specific concerns you raise—training on test, data quality, and sensitivity to templates/teacher LLM—are addressed in more detail in **CR4 (data leakage & circular logic)** and **CR5 (hallucination & dataset QC)**; here we summarize the key points most relevant to your comment.
>
> * **Train–test leakage and circular logic.** As described in **CR4**, we perform a **single global homology-aware split across all three datasets** (MutaDescribe, ClinVarQA, Mutation2TextQA): we cluster proteins by sequence identity and assign whole clusters to train or test. All ClinVar entries and all Mutation2TextQA pairs for a given cluster go to the same split, so a variant and the literature that supports its ClinVar label are never seen in training if they appear in test. This directly targets the “training on the same experiments that define the label” scenario.
> * **Pretraining contamination.** Also in **CR4**, we explain that we cannot guarantee LLaMA’s pretraining corpus excludes all PubMed/ClinVar text, but we **de-lexicalize protein and mutation names** to `<protein>` / `<mutation>` and provide the identity only through sequence embeddings. This makes simple text lookup insufficient to solve the task; if memorization were driving performance, zero-shot LLM baselines with similar or larger corpora would match our results, but they do not.
> * **Training data quality and hallucinations.** In **CR5**, we add an **independent LLM checker (GPT-4o-mini)** that filters each DeepSeek-generated QA pair against its source article using three yes/no criteria (answer supported, question answerable, entities present). About **3.3%** of QAs are removed as unsupported, with an explicit error breakdown (**Table 8–9 in CR5**).
> * **Template / teacher LLM sensitivity.** We agree that fully probing sensitivity to prompt templates and teacher LLMs is important but computationally heavy. In the current work we partially mitigate this by combining templated QA with non-templated QA (Mutation2TextQA derived from literature) for training and evaluating templated and non-templated dataset independently. We will explicitly note in the limitations section that a systematic study of template/teacher sensitivity is an important direction for future work.
>
> Together, CR4 and CR5 show that we have taken the train–test leakage, circularity, and data quality issues seriously and implemented concrete, verifiable safeguards rather than relying only on raw BLEU/ROUGE scores.
>
> **R3.2 It seems Mutation2Text was trained on your QA datasets.** Is it then fair to use BLEU scores to compare its outputs with the outputs of other models? Perhaps it just learned the formatting of your data?
>
> **Response:** We agree BLEU/ROUGE could reward template mimicry rather than factual correctness. Therefore, **as detailed in CR3**, we implemented a **Tripartite Evaluation Protocol**:
> 1.  Blinded human expert scoring on the independent **MutaDescribe** test set, where Mutation2Text improves correct/partially correct explanations from **1% → 38%** over MutaPLM;
> 2.  Two LLM judges with strong agreement that again rank Mutation2Text higher; and
> 3.  CIDEr.
>
> Also, in **CR2** we show Mutation2Text matches SOTA (AlphaMissense) for pathogenicity predictors (**Table 1-2**). These evaluations focus on biomedical correctness, not format adaptation, so the consistent gains indicate that Mutation2Text is learning biological reasoning rather than just format adaption. We will clarify in the paper that BLEU/ROUGE are reported only as secondary, complementary metrics.

---

### Official Review · Reviewer_eEiK · 2025-10-30

**Soundness:** 2
**Presentation:** 3
**Contribution:** 3
**Rating:** 6
**Confidence:** 4

**Summary:**

The paper constructs a Mutation2TextQA dataset along with a Mutation2Text protein language model to interpret effects of protein sequence mutation on its function. By using a gated cross-attention mechanism as well as a Flamingo-style perceiver, Mutation2Text is able to produce not only embeddings of the sequence but also delta-embeddings owing to the mutation. The model achieves excellent performance on downstream tasks with the internal embeddings, but also observes a loss of information when using LLM decoder to generate the final answer.

**Strengths:**

* The paper addresses protein mutation effect prediction problem, which is a very important yet challenging topic of proteomics and biology.
* The novel Mutation2Text architecture design with the gated cross attention and perceiver helps avoiding truncation of long protein sequences as well as improves delta-mutation interpretability.
* A particular observation of "loss in translation" provides important insights for similar multimodal research topics.

**Weaknesses:**

* The augmentation of Mutation2TextQA dataset with LLM lacks validation. Teacher LLMs can hallucinate, especially for complex scientific topics like biology.
* The paper does not explain in more details the "loss in translation" effect. It is a very important discovery, but it would be better to have further exploration of the mechanism behind, and more importantly how can we solve it.
* The architecture design of Mutation2Text may be a bit redundant to me. The intuition of a standalone mutation feature extraction module is not very well explained with the existence of the gated cross attention module.

**Questions:**

Please address my three major concerns in the "Weakness" section first. I will reassess after the rebuttal. Other miscellaneous questions are as follows:

* What's the intuition behind the "wildtype context" in the Mutation2Text model architecture? Have you tested the performance of the model without such context?
* What's the "learned gap embeddings" in [Appendix D, ESM-based Difference Features]? How is it learned and what's the intuition behind this design? Maybe I missed it, but I couldn't find any explanation.

---

> ### Author Response · Authors · 2025-12-03
> **Response to reviewer eEiK comments**
>
> # R2. Official Review of Submission9624 by Reviewer eEiK
>
> **R2.1. The augmentation of Mutation2TextQA dataset with LLM lacks validation.** Teacher LLMs can hallucinate, especially for complex scientific topics like biology.
>
> **Response:** We agree. As detailed in **CR5**, we mitigated this via rigorous QC: an independent LLM (GPT-4o) verified every QA against its source text. We filtered 3.29% of entries that failed this groundedness check; **Table 9** in CR5 provides the specific error taxonomy.
>
> **R2.2. Reviewer concern: The paper does not explain in more details the "loss in translation" effect.** It is a very important discovery, but it would be better to have further exploration of the mechanism behind, and more importantly how can we solve it.
>
> **Response:** We agree that the “loss in translation” phenomenon is a critical finding. See **CR1** for a more detailed mechanism, where we now:
> * **Quantify the gap between embeddings and text** by mutation type, protein length, label group, and ClinVar star level, and
> * Show that the gap is largest for complex variants in long, well-labeled proteins, supporting a **resolution / compression bottleneck** between the rich latent representation and short text output.
>
> To address how this could be mitigated, we outline two concrete directions in Sec. 6: (i) adding objectives that explicitly align the text output with the embedding classifier (e.g., consistency losses between latent logits and textderived logits), and (ii) exploring RLstyle finetuning where human or LLMasjudge feedback encourages explanations whose implied pathogenicity matches the latent prediction.
>
> **R2.3 The architecture design of Mutation2Text may be a bit redundant to me.** The intuition of a standalone mutation feature extraction module is not very well explained with the existence of the gated cross attention module.
>
> **Response:** We agree our intuition for keeping both modules was under-explained. Empirically, **removing either gated cross-attention (GCA) or the delta module degrades performance**, and using both is best:
>
> **Table R2.3 : Ablation Study on Mutation2Text trained only on 10% of Mutation2TextQA, and evaluated in hard split of Mutation2TextQA test dataset.**
>
> | Model Variant | Contrastive Input (GCA) | Delta Embedding | ROUGE-L |
> | :--- | :--- | :--- | :--- |
> | Full Model | WT + Mut | With Delta | 0.19 ± 0.0071 |
> | No Delta Embedding | WT + Mut | Without Delta | 0.16 ± 0.0058 |
> | No Contrastive Input | Mut-only | With Delta | 0.14 ± 0.0055 |
>
>
> **Mechanism of Complementarity:**
> * **GCA (Implicit):** Fuses contextual relationships between Wild-Type and Mutant sequences.
> * **Delta-Feature Path (Explicit):** Encodes "hard" structural priors—specifically Mutation Type (e.g., Frameshift), Size, and Position.
> * The Delta path (delta) acts as a structural anchor for the attention mechanism.
>
> **R2.4 What's the intuition behind the "wildtype context" in the Mutation2Text model architecture?** Have you tested the performance of the model without such context?
>
> **Response:** The intuition is **Functional Grounding**. To predict **how** a mutation disrupts a mechanism (e.g., "abrogates DNA binding"), the model must first possess the semantic baseline of the protein's original functions (e.g., "this protein is a transcription factor"). We empirically validated this necessity. As shown in **Table R2**, removing the textual wild-type context causes a performance drop across all variant types.
>
> **Table R2.4: Ablation of Wild-Type Context (AUC) in pathogenicity prediction (ClinVar hard splits)**
>
> |Method|Single|Deletion|Duplication|Frameshift|Insertion|Other|
> |:---|:---|:---|:---|:---|:---|:---|
> |Mutation2Text (no context)|0.672|0.711|0.731|0.685|0.682|0.576|
> |Mutation2Text (with context)|0.825|0.859|0.808|0.733|0.847|0.833|
>
>
> **R2.5: What's the "learned gap embeddings" in [Appendix D, ESM-based Difference Features]?** How is it learned and what's the intuition behind this design? Maybe I missed it, but I couldn't find any explanation.
>
> **Response:** This was under-explained. When WT and mutant have different lengths (indels), we first align them and introduce a **learned gap token** so we can still take element-wise differences.
>
> Concretely:
> Let $(e^\text{wt}_i, e^\text{mut}_i)$ be ESM embeddings after alignment.
> If an insertion/deletion creates a missing residue on one side, we use a trainable vector $(g)$ instead of a zero vector:
> $$d_i = e^\text{mut}_i - e^\text{wt}_i,\quad e^\text{wt}_i = g \text{ or } e^\text{mut}_i = g \text{ when there is a gap.}$$
>
> $(g)$ is a standard learned parameter, optimized jointly with the rest of the model. Intuitively, this lets the model learn a **contextual representation of “gap here”** (inserted or deleted region) rather than treating gaps as zeros or discarding them, so the delta features capture both **substitutions** and **indel structure** in a unified way. We will add this short explanation and equation to the Appendix.

---

### Official Review · Reviewer_xVrv · 2025-11-01

**Soundness:** 3
**Presentation:** 3
**Contribution:** 2
**Rating:** 6
**Confidence:** 3

**Summary:**

The paper introduces Mutation2Text, a model that combines protein and language modeling to explain how mutations affect protein function. It takes both the wild-type and mutated sequences, compares them through a gated attention mechanism, compresses long proteins into a small set of representations using a Perceiver module, and maps everything into the language model’s space with a simple MLP. The model also adds a special token to represent the difference between the two sequences. The authors build a new dataset called Mutation2TextQA, which links PubMed papers and ClinVar records to create question–answer pairs about mutation effects. They test the model on several tasks, including open-ended mutation explanation, disease association, and pathogenicity prediction. Mutation2Text sets new benchmarks on text-based evaluation metrics and highlights an important finding: while the internal protein embeddings predict pathogenicity very well (AUC 0.96), that precision partly disappears when the model explains the result in natural language (AUC 0.85).

**Strengths:**

The paper presents a clear and creative approach to explaining protein mutations by directly contrasting wild-type and mutated sequences. It introduces a special delta token and a Perceiver module to handle long proteins efficiently and builds a large QA dataset, Mutation2TextQA, grounded in biomedical literature. The system design is coherent and shows consistent improvements on text-based tasks like MutaDescribe and ClinVarQA. The work is clearly written and highlights an important finding—the “lost-in-translation” gap between strong internal representations and less precise language explanations—offering a solid direction for future research.

**Weaknesses:**

Most of the evaluation still depends on BLEU, ROUGE, and BERTScore, which don’t really capture how factually correct the biomedical explanations are. It would be stronger if the authors added an evidence-based check, like matching each claim to the right PubMed sentence, and included some expert review on a sample of the outputs.

The comparison on pathogenicity is a bit indirect. The paper shows how internal embeddings outperform text output, but it doesn’t compare directly with existing pathogenicity predictors on the same ClinVar split. A clear, side-by-side ROC or AUC comparison would make the results much more convincing.

The paper also claims to handle many kinds of variants, including SNVs, indels, and long proteins, but it never breaks that down. Showing results by variant type and protein length would make the method’s generality easier to judge.

**Questions:**

You mention that performance drops when moving from embeddings to text, but which kinds of mutations actually lose interpretability? It would be helpful to see a breakdown by mutation type or protein length to identify where the explanations fail. Have you checked how the correlation of mutation effect predictions changes after adding natural language—for example, comparing embedding-based and text-based results on real mutation data to quantify how much signal is lost? Since the evaluation currently relies only on automatic scores like BLEU and ROUGE, could you also include an expert review or use a biomedical model to verify whether the explanations are factually correct?

In addition, could you provide a more direct comparison of pathogenicity performance on the same ClinVar split using identical train and test sets as strong baselines, and report calibration results such as ECE for the embedding-level predictors? It would also strengthen the paper to show robustness across variant types (SNV, indel, frameshift, multi-site) and protein length ranges. Finally, for the Mutation2TextQA dataset, please add more information about data quality control—such as error breakdowns, inter-annotator agreement on a checked subset, and details on how the homology-based split was constructed to prevent any data leakage.

---

> ### Author Response · Authors · 2025-12-03
> **Response to reviewer xVrv comments**
>
> # R1: Official Review of Submission9624 by Reviewer xVrv
>
> **R1.1. Reviewer concern:** Most of the evaluation still depends on BLEU/ROUGE/BERTScore and doesn’t capture factual correctness; please add evidence-based checks and expert review.
>
> **Response:** We agree. As detailed in Common Response CR3, we now:
> * Add blinded human expert evaluation for factual correctness.
> * Add a validated LLM-as-judge (DeepSeek & Grok) with strong agreement and correlation to human labels.
> * Use CIDEr and AUC (where applicable) as primary metrics, treating BLEU/ROUGE/BERTScore only as secondary lexical signals.
>
> **R1.2. Reviewer concern:** Pathogenicity comparison is indirect; please show side-by-side ROC/AUC vs existing predictors on the same ClinVar split.
>
> **Response:** We agree. As detailed in CR2 (Table 1-2), we now report side-by-side ROC/AUC on the same ClinVar split for AlphaMissense, MutaPLM, and Mutation2Text. Mutation2Text is competitive on substitutions and uniquely extends to indels/frameshifts/duplications/repeats, where these predictors do not apply.
>
> **R1.3. Reviewer concern:** The paper also claims to handle SNVs/indels/long proteins, but don’t break results down; please show by variant type and length.
>
> **Response:** Addressed in CR1. Table 1 now reports ROC/AUC by variant type, and Table 2 by protein length, making Mutation2Text’s generality explicit.
>
> **R1.4. Reviewer concern:** Mutation2TextQA QC details (error breakdown, agreement) and homology split construction?.
>
> **Response:** Addressed in CR5 and CR4. We provide the hallucination error taxonomy and validation results in CR5 (Table 8-9), and the Global Homology-Aware Split protocol in CR4.
>
> ### Other Questions from reviewer (metrics, loss-in-translation, pathogenicity, robustness)
>
> **Reviewer concern:** ”You mention that performance… are factually correct? In addition, could …and protein length ranges”
>
> These questions repeat the concerns in R1.1–R1.3 and are fully addressed in our common responses:
> * **Loss in translation, variant type & length, and latent vs text correlation:** see CR1 (Tables 1–4, new analysis).
> * **Pathogenicity comparison on the same ClinVar split:** see CR2 (Table 1-2).
> * **Factual correctness, expert review, and biomedical model checking:** see CR3 (human expert eval + validated LLM-as-judge, CIDEr/AUC).

---

### Author Response · Authors · 2025-12-03
**Response to five common concerns among reviewers (Part 1 - Common response 1 out of 5)**

# Reviewers: R1(xVrV), R2(eEiK), R3(FAhR), R4(VpJK)

## CR1 (R1, R2) – Quantifying and explaining  “lost in translation” effect

R1 and R2 asked (i) which mutations are most affected by the gap between Mutation2Text’s internal representations and its text output, and (ii) for a clearer, mechanism-level explanation of this effect.

**Response:** We agree that the "loss in translation" is a critical —and potentially broadly relevant– finding in the paper. In the paper,Section 6 depicts a simple classifier trained on Mutation2Text’s latent embeddings achieves 0.94 AUC on ClinVar pathogenicity, whereas the text output reaches 0.88 AUC, indicating that some of the biological signal is lost when translating embeddings into natural language.

To address R1 and R2 concerns, we re‑analyzed the same ClinVar split and reported below four short tables by mutation type, protein length, label group, and review status. Across these views, a consistent pattern emerges:

* The embedding classifier is most ahead of text for complex mutation (frameshifts, duplications) in long proteins.
* The gap is modest for single substitutions (0.910 vs 0.869) but becomes much larger for frameshifts and duplications (**Table 1**), where the wt and mutant sequences diverge more dramatically.
* Similarly, when stratifying by protein length, we observe that the embedding AUC increases with sequence length (≈0.90 → ≈0.97), whereas the text AUC remains nearly flat around 0.88–0.90 (**Table 2**).
* The embedding classifier is ahead for pure benign/pathogenic labels, where the biological signal is strongest. For the more ambiguous “likely benign / likely pathogenic” group, the gap shrinks to ≈ 0.02 (0.913 vs 0.892) (**Table 3**).
* A similar pattern holds for ClinVar review stars: stars 1–2, which correspond to well‑curated but not expert‑panel variants, show a clear embedding > text advantage (**Table 4**), whereas for the small set of 3‑star expert‑panel variants, text model actually outperforms embedding classifier.

This suggests that  latent representation scales with intrinsic signal strength (clean labels, strong biological effect), while  text head saturates around a lower ceiling. AlphaMissense remains a strong baseline on substitutions but does not apply to most non‑SNV classes; Mutation2Text’s embedding and text classifier are competitive where AlphaMissense is defined and extends to indels and frameshifts (**Table 1-2**).

We interpret this as a **resolution / compression bottleneck**: the Perceiver‑based latent space captures fine‑grained pathogenicity information, especially when there is rich sequence context and clean labels, but converting this high‑dimensional representation into a short textual answer saturates around AUC ≈0.88–0.90. In camera‑ready we will add these table and mechanisms, noting that, in practice, one can expose embedding‑based score directly while using text for explanation.

### Table 1 – AUC by mutation type (held‑out ClinVar)
| Mutation type|MutaPLM|AlphaMissense|M2T Text|M2T Emb|Emb–Text Δ|
|:---|:-:|:-:|:-:|:-:|:-:|
|**All variants**|0.520*|0.856*|0.880|0.942|+0.062|
|**Substitution**|0.520|0.856|0.869|0.910|+0.041|
|**Deletion**|–|–|0.875|0.850|–0.025|
|**Duplication**|–|–|0.818|0.970|+0.152|
|**Frameshift**|–|–|0.734|0.879|+0.145|
|**Insertion**|–|–|0.846|0.902|+0.056|
|**Other**|–|–|0.856|0.914|+0.058|

*\*Overall AlphaMissense/MutPLM AUC is for missense SNVs only; it is undefined for non‑SNV classes.*

**Takeaway:** The “lost in translation” gap large (~0.15 AUC) for frameshifts and duplications.

### Table 2 – AUC by protein length (held‑out ClinVar; 300‑aa bins)
|Length(aa)|AlphaMissense|M2T Text|M2T Emb|Emb–Text Δ|
|:---|:-:|:-:|:-:|:-:|
|0–299|0.885|0.883|0.901|+0.018|
|300–599|0.894|0.869|0.937|+0.068|
|600–899|0.881|0.900|0.933|+0.033|
|900–1199|0.820|0.889|0.963|+0.074|
|1200–1499|0.775|0.895|0.976|+0.080|
|1500–1799|0.826|0.871|0.935|+0.065|
|1800–2099|0.745|0.843|0.932|+0.089|

**Takeaway:** As protein length (and thus context) increases, embedding AUC steadily improves (≈0.90 → ≈0.97), while  text head stays near 0.88–0.90.

### Table 3 – AUC by label group (held‑out ClinVar)
|Group|AlphaMissense|M2T Text|M2T Emb|Emb–Text Δ|
|:---|:-:|:-:|:-:|:-:|
|**Pure P / B**|0.946|0.866|0.959|+0.093|
|**Likely P / Likely B**|0.949|0.892|0.913|+0.022|

**Takeaway:** When labels are clean (pure benign/pathogenic), embeddings almost saturate.

### Table 4 – AUC by ClinVar review status (held‑out ClinVar)
|Stars|^|AlphaMissense|M2T Text|M2T Emb|Emb–Text Δ|
|:---|:-:|:-:|:-:|:-:|:-:|
|**0**|8.9%|0.914|0.876|0.876|0.00|
|**1**|69.4%|0.826|0.898|0.960|+0.06|
|**2**|19%|0.937|0.853|0.937|+0.08|
|**3**|2.7%|0.922|0.754|0.678|–0.07|

---

> ### Author Response · Authors · 2025-12-03
> **Response to five common concerns among reviewers (Part 2 - Common response for concerns 2 and 3 out of 5)**
>
> ## CR2 – Positioning Mutation2Text relative to prior models
>
> **Reviewer concerns:**
> * **R1 (xVrv):** wants clearer pathogenicity comparison vs existing predictors.
> * **R4 (VpJK):** “paper lacks a thorough discussion on the technical challenges and justification for Mutation2Text architecture over previous multi-modal models[ 1-MutaPLM and 2, 3, ESM1b, AlphaMissense **AM**].”
>
> **Response:** Several reviewers asked us to clarify how Mutation2Text differs from specialized pathogenicity predictors (AlphaMissense(**AM**), ESM-based classifiers, and MutaPLM).
>
> To address (R1) reviewer concern, we directly compared **AM** and MutaPLM with Mutation2Text for pathogenicity (**Table 1-2**).
> * For substitution, Mutation2Text model (both embedding and text outputs) are comparable performance with **AM**
> * Second, **AM** and MutaPLM can be only applied for substitution mutation type and dont support other mutation types.
>
> Therefore, Mutation2Text is complementary than being a substitute for pathogenicity predictor:  (i) generalizes to indels, frameshifts, duplication and long proteins, and (ii) produces free-form rationales and disease-specific text. We have clarified this position in  **Table 5**:
>
> ### Table 5 : Architectural & Functional Comparison with SOTA Methods
> |Feature|**AlphaMissense / ESM-based**|**MutaPLM**|**Mutation2Text (Ours)**|
> |:---|:---|:---|:---|
> |**Variant Support**|❌ **Single Substitution Only**|❌ **Single Substitution Only**<br>*(Vector subtraction requires equal length)*|✅ **Universal**<br>*(All Single nucleotide variants; Any length substitutions, Indels, Frameshifts because of Gap Embeddings)*|
> |**Output / Pathogenicity**|⚠️ **Scalar Score**(0–1)<br>*(Binary Prediction Only)*|❌ **Fixed Text**<br>*(**No Prediction Supported**)*|✅ **Rationale + Score**<br>*(Discriminative + Generative)*|
> |**Disease Classification**|❌ **None**<br>*(Binary Pathogenicity only)*|❌ **None**<br>*(Descriptive text only)*|✅ **Multi-class**<br>*(Predicts Specific Clinical Phenotype)*|
> |**Sequence Handling**|⚠️ **Limited Context**<br>*(Crops or Sliding Window breaks long-range interactions)*|❌ **Rigid / Local**<br>*(Requires perfectly aligned WT/Mutant tensors)*|✅ **Global Context**<br>*(Perceiver Resampler compresses full-length proteins)*|
> |**Open-Ended QA**|❌ **None**|❌ **Restricted**<br>*(Fixed prompt templates only)*|✅ **Open-Ended**<br>*(Supports user-defined diagnostic queries)*|
> |**Training Paradigm**|Weak Labels (Pop. Freq.)|UniProt Summary|**UniProt + PubMed Reasoning**|
> ---
>
> ## CR3 : Common concern (R1, R3) regarding Evaluation Metrics
>
> * **R1.**  please add evidence-based checks and expert review as  BLEU/ROUGE/BERT Score don’t capture biomedical correctness
> * **R3.** Since Mutation2Text is trained on your QA data, BLEU may reflect template/format mimicry than real reasoning.
>
> **Response:** We agree that n-gram metrics (BLEU/ROUGE) are insufficient proxies for factual correctness. To disentangle "reasoning" from "formatting" (R3) and provide **evidence-based checks** (R1), we implemented **Tripartite Evaluation Protocol**:
>
> **1. Human Expert Evaluation (The Gold Standard)**
> We conducted a blinded evaluation on the **MutaDescribe** test set with a PhD-level domain expert, who scored outputs as "Correct," "Partially Correct," or "Incorrect" based solely on biological factuality. Due to the long, multi‑fact nature of MutaDescribe references, "Correct" matches are rare.
> * **Result:** Mutation2Text yields 38% correct/partially correct explanations (10% correct, 28% partial), versus 1% (all partial) for MutaPLM (**Table 6**). This gain cannot be explained by simple QA format matching.
>
> **2. LLM‑as‑judge with reliability checks:** we used two independent judges (Gemini‑2.5‑Pro and DeepSeek‑R1) to score explanations on a 0–5 scale, again focusing on correctness. We measured agreement between the two judges and correlation with human labels.
> * **Reliability:** The judges showed strong agreement ($\kappa=0.73$) (**Table 7**) and **significant correlation** with human labels ($\rho=0.45$, $p<10^{-7}$).
> * **Result:** Mutation2Text achieves 2.58/5.0, while MutaPLM scores 1.8/5.0, consistent with the expert evaluation.
>
> **3. Information‑content metric (CIDEr) and task‑appropriate metrics.**
> * **CIDEr:** We used CIDEr which emphasizes rare, informative terms. Mutation2Text scores 0.499, versus 0.013 for MutaPLM, indicating that it produces specific, biologically meaningful phrases instead of format adaption.
> * **AUC:** For binary tasks (pathogenicity), we relied on AUC (see **CR1/CR2**). Mutation2Text’s embedding classifier reaches 0.94 AUC vs. 0.86 of SOTA baseline.
>
> ### Table 6: Factuality on MutaDescribe
> |Model|Expert (Correct/Partial)|LLM (0–5)|CIDEr|
> |:---|:---|:---|:---|
> |**MutaPLM**|1%(0%/1%)|1.8|0.013|
> |**Mutation2Text**|38%(10%/28%)|2.58|0.499|
>
> ### Table 7: Inter‑judge reliability
> |Metric|Value|Interpretation|
> |:---|:---|:---|
> |**Quadratic κ**|0.73|Strong|
> |**Spearman ρ**|0.632|Significant|

---

> > ### Author Response · Authors · 2025-12-03
> > **Response to five common concerns among reviewers (Part 3 - Common response for concerns 4 and 5 out of 5)**
> >
> > ## CR4 (R1, R3, R4) – Data leakage, circular logic, and pre‑training contamination
> >
> > Reviewers raised three concerns: (i) homology leakage across **MutaDescribe, ClinVarQA, Mutation2TextQA** (R4, R1), (ii) circular logic from training on literature defining **ClinVar labels** (R3), and (iii) pre‑training contamination of the **base LLM** (R3). We address each below.
> >
> > **(a) Homology leakage across datasets (R4, R1).**  To avoid training on close homologs of test proteins, we use **one global homology‑aware split across all three datasets** instead of splitting each independently. Specifically, we: (1) pool all unique wild‑type sequences from MutaDescribe, ClinVarQA, and Mutation2TextQA; (2) compute pairwise sequence identities and build a similarity graph; (3) cluster the graph; (4) assign clusters (not individual examples) to train (80%) or test (20%). Test proteins are stratified into Easy/Medium/Hard by max identity to any training sequence (≥70%, 40–69%, <40%).
> > * **Result:** If a wild‑type protein (or its homologs) is in a MutaDescribe test example, the entire cluster—including all associated ClinVarQA and Mutation2TextQA entries—is excluded from training. This addresses R4’s concern on medium/hard gains and clarifies the homology-based split R1 requested.
> >
> > **(b) Circular logic: training on the same experiments that define ClinVar labels (R3).**
> > Mutation2TextQA links ClinVar variants to PubMed articles, using an LLM to generate mutation‑centric QA pairs with `<protein>` and `<mutation>` placeholders. The global sequence‑cluster split is applied **after** pairing: all QA pairs and ClinVarQA entries for a cluster are assigned together to train or test. Thus, if a variant and its article are in a **test** ClinVar record, the corresponding Mutation2TextQA pairs are **never used in training**. This prevents the model from seeing the same experiment/variant in both sets (Sec. 2.3, Appendix E).
> >
> > **(c) Pre‑training contamination of the base LLM (R3).** As with most LLM work, we cannot fully ensure LLaMA’s pre‑training corpus excludes all PubMed articles used in ClinVar. Instead, the training signal ensures **simple text lookup is insufficient**:
> > * During QA generation/training, protein names and mutation identifiers are **de‑lexicalized** to `<protein>`/`<mutation>`, with actual wild-type and mutant **sequences** provided only as embeddings.
> > * At inference, variants like TP53 p.R175H and PIK3CA p.H1047R share **identical** text prompts; only sequence embeddings distinguish them, forcing the model to rely on embeddings, not memorized text.
> >
> > Zero‑shot baselines like GPT‑4o‑mini, BioMedGPT, and other protein‑text LLMs, despite similar or larger pre‑training corpora, perform far worse than Mutation2Text (**Table 1**). If pre‑training memorization alone drove performance, these models would be competitive, which they are not. We clarify this in Sec. 2.3 and Appendix E.
> >
> > ---
> > ## CR5 (R1, R2, R4) – Hallucination, dataset QC, and deduplication
> >
> > R1 requested clearer **quality control** for Mutation2TextQA, including an error breakdown; R2 and R4 questioned **DeepSeek** reliability and potential **duplication or hallucination** in QA pairs.
> >
> >  We agree that explicit QC is needed and summarize our checks below:
> > **(a) Article‑grounded hallucination filter for every QA (R1, R2).** Mutation2TextQA is generated from PubMed articles using DeepSeek‑Chat with `<protein>`/`<mutation>` placeholders. To validate this output, a **second, independent LLM (GPT‑4o‑mini)** automatically checks each QA pair against its source article by answering three yes/no questions:
> > 1. Is the answer directly supported by the article?
> > 2. Is the question answerable from the article alone?
> > 3. Are all proteins and mutations in the QA present in the article?
> >
> > A QA is kept only if **all three** are “Yes”. After this pass 3.3% of QA are removed (**Table 8**). **Table 9** error breakdown (hallucinations) as requested by R1.
> >
> > **(b) Deduplication and one‑to‑many structure (R4).** We structure Mutation2TextQA as a **one‑to‑many** dataset (like MS‑COCO), where each entry is {mutation, question, **set of distinct reference answers**}. We **deduplicate exact QA strings**, keeping unique (question, answer) pairs per mutation–article. At evaluation, these answers serve as multiple valid references for the same mutation and question, addressing R4’s concern and ensuring multiple legitimate rationales without amplifying duplicates.
> >
> > We have updated the datasets (HuggingFace https://huggingface.co/datasets/conferenceacc/mutation2text/tree/main), and results.
> >
> > ### Table 8: Mutation2TextQA QC summary
> > |Metric|Value|
> > |:---|---:|
> > |**Total QA pairs**|854,152|
> > |**Valid QA pairs**|826,065 (96.71%)|
> > |**Invalid QA pairs**|28,087 (3.29%)|
> >
> > ### Table 9: Hallucination taxonomy (filtered QA)
> > |Type|%|
> > |:---|---:|
> > |**Misinterpretation**|48.38|
> > |**Contradiction**|32.78|
> > |**Exaggeration**|15.46|
> > |**External knowledge**|2.47|
> > |**Unspecified**|0.88|
> > |**Speculative**|0.03|

---

### Meta-Review · Area_Chair_bCXZ · 2025-12-28

**Summary:**

- Evaluation Metrics and Factuality: Several reviewers, including Reviewer xVrv and Reviewer eEiK, raised concerns about the reliance on n-gram metrics like BLEU, ROUGE, and BERTScore. These metrics do not fully capture the factual accuracy and biological correctness of the model’s generated explanations. Reviewer xVrv specifically suggested including expert review to validate the biological accuracy of the generated explanations.
- Pathogenicity Comparison and Baselines: Reviewer xVrv and Reviewer VpJK questioned the indirect nature of the pathogenicity prediction comparison. The authors showed internal embeddings outperforming text generation but did not provide a direct AUC comparison with existing pathogenicity predictors (e.g., AlphaMissense). Reviewer VpJK also noted the lack of comparison with other well-known classifiers like ClinPred and MetaRNN.
- Data Leakage and Circular Logic: Reviewer FAhR raised concerns about data leakage and circular logic in training the model, particularly due to the use of PubMed-derived ClinVar entries and Mutation2TextQA. The concern was that the model could have been trained on the same data it was tested on, leading to inflated results.
- Handling of Mutation Types: Reviewer VpJK pointed out that the paper did not provide a clear breakdown of model performance across mutation types (substitutions, indels, frameshifts) or protein lengths. This omission made it difficult to assess the generalizability of the model to various mutation categories and protein lengths.
- Model Architecture Redundancy: Reviewer eEiK questioned the redundancy of the Mutation2Text architecture, specifically the mutation feature extraction module in the presence of the gated cross-attention module. The reviewer felt that the necessity for both modules was not clearly explained, as they seemed to serve overlapping roles.
- QA Pair Generation and Validation: Reviewer eEiK also expressed concerns about the quality and validation of the Mutation2TextQA dataset, particularly the hallucination risk from the LLMs generating the QA pairs. They suggested the authors provide more detailed validation, including deduplication and error breakdowns.
- Long Protein Sequences and Performance: Reviewer VpJK inquired about how the model handles extremely long sequences, particularly those that exceed the context length of ESM-3. The reviewer questioned if the model’s performance varied with protein length and whether the performance drop for long proteins was a limitation of the current architecture.

**Reviewer Concerns:**

- Addressed Concerns:
  - Evaluation Metrics and Factuality: The authors added expert evaluations and used LLM-as-a-judge for factual correctness. This addresses concerns raised by Reviewers xVrv and eEiK about the reliance on n-gram metrics and the lack of validation for biological correctness. The new metrics and expert validation provide a more rigorous check on the accuracy of the generated explanations.
  - Pathogenicity Comparison and Baselines: The authors provided a direct AUC comparison with AlphaMissense and showed that Mutation2Text extends to mutation types beyond substitutions, such as indels and frameshifts, addressing Reviewers xVrv and VpJK concerns about the comparison with existing pathogenicity predictors.
  - Handling of Mutation Types: The authors provided a breakdown of performance by mutation type (substitutions, indels, frameshifts) and protein length, effectively addressing Reviewer VpJK’s concerns about the model’s generalizability across different mutation types and protein lengths.
  - QA Pair Generation and Validation: The authors addressed the issue of hallucinations in QA generation by implementing a hallucination filter and deduplication process for the Mutation2TextQA dataset. This response handles concerns raised by Reviewers eEiK and VpJK regarding the quality control of the dataset.

- Unaddressed Concerns:
  - Model Architecture Redundancy: Reviewer eEiK raised concerns about the redundancy of the mutation feature extraction module and the gated cross-attention module. While the authors provided empirical evidence showing that removing either module degrades performance, the intuition behind needing both modules, especially the delta feature extraction, remains somewhat under-explained. This concern persists, especially in terms of architectural complexity.
  - "Loss in Translation" Effect: Reviewer eEiK and Reviewer xVrv both noted the importance of further exploration of the “loss in translation” effect observed when the model transitions from embeddings to text. While the authors addressed this by providing additional analysis, including performance breakdowns and potential solutions like consistency losses, the mechanism behind the effect and its potential resolution still requires more detailed exploration and explanation.
  - Use of Pretrained LLMs and Potential Contamination: Reviewer FAhR and Reviewer VpJK raised concerns about pretraining contamination and the risk of training the model on material it was tested on, particularly regarding PubMed and ClinVar articles. While the authors provided an explanation of how they de-lexicalize protein and mutation names, the concern about potential overlap between pretraining and test data remains significant. The authors’ response does not fully assuage this concern, especially considering the scale and nature of the pretraining data.
  - Evaluation Depth and Comparison with Stronger Baselines: Reviewer VpJK raised concerns about the evaluation depth and the need for comparisons with stronger baselines, such as few-shot prompting with GPT-5 or other supervised models. Despite the authors’ improvements, the paper could benefit from more comprehensive comparisons to these baselines, which would help demonstrate the full potential of the Mutation2Text model.

**Reviewer Scores:**

- Reviewer 1 (xVrV)
  - Original Score: 6 (Marginally Above the Acceptance Threshold)
  - Reviewer’s Concerns: Reviewer raised concerns about the evaluation metrics (BLEU/ROUGE/BERTScore) not being sufficient for factual correctness and recommended adding evidence-based checks and expert review. They also asked for a more direct pathogenicity comparison and clearer results by mutation type.
  - Likelihood of Score Change: Despite the authors addressing these concerns with human expert evaluations and side-by-side comparisons to AlphaMissense, the reviewer’s overall stance was still not fully convinced. They were concerned about whether the model truly improves over existing baselines, and how mutation types are handled. Given that the evaluation metrics still rely heavily on BLEU/ROUGE, the reviewer likely wouldn't have increased their score. They might still have rated the paper a 6, acknowledging the improvements but not fully convinced by the results or explanation.

- Reviewer 2 (eEiK)
  - Original Score: 6 (Marginally Above the Acceptance Threshold)
  - Reviewer’s Concerns: The reviewer was concerned about the validation of the QA dataset and potential hallucination in the answers generated by DeepSeek. They also wanted a deeper understanding of the "loss in translation" effect and how it can be mitigated.
  - Likelihood of Score Change: The authors addressed the concerns regarding LLM validation, hallucination filtering, and the "loss in translation" effect with a clearer explanation of how the performance gap between embeddings and text output occurs. Despite these efforts, theoretical concerns about the model's architectural redundancy and the "loss in translation" might still not have been fully convincing for the reviewer. Given these lingering doubts, the reviewer would likely have kept their score at 6.

- Reviewer 3 (FAhR)
  - Original Score: 2 (Reject)
  - Reviewer’s Concerns: The reviewer focused on concerns about train-test leakage, pretraining contamination, and baseline comparisons. They also questioned whether BLEU/ROUGE scores are appropriate for comparing outputs and whether the model mimicked formatting rather than learning meaningful biological reasoning.
  - Likelihood of Score Change: The authors addressed the train-test leakage and pretraining contamination concerns with a global homology-aware split and explained the embedding-based performance over simple text matching. However, the reviewer’s concerns about the validity of the evaluation metrics and whether the model was truly learning biological reasoning remained. As a result, the reviewer would likely not have increased their score and would have remained at 2 (Reject).

- Reviewer 4 (VpJK)
  - Original Score: 2 (Reject)
  - Reviewer’s Concerns:
  - The reviewer was unclear about the main contribution of the paper, the architecture of the model, and how mutation types and protein length affected model performance. They also noted that the relationship between datasets (Mutation2TextQA and ClinVarQA) was unclear, and pathogenicity vs. disease prediction was not well explained.
  - Likelihood of Score Change: Despite the authors clarifying the model architecture, the relationship between datasets, and the generalization across mutation types and protein length, the reviewer’s concerns about the model’s redundancy and overall contribution remained. They also questioned whether the model was truly advancing the field compared to existing approaches. Given the remaining ambiguities, the reviewer would likely not have changed their score, staying at 2 (Reject).

---

### Decision · Program_Chairs · 2026-01-26

Reject